# Botulinum Toxin Therapy for Oromandibular Dystonia and Other Movement Disorders in the Stomatognathic System

**DOI:** 10.3390/toxins14040282

**Published:** 2022-04-14

**Authors:** Kazuya Yoshida

**Affiliations:** Department of Oral and Maxillofacial Surgery, National Hospital Organization, Kyoto Medical Center, 1-1 Mukaihata-cho, Fukakusa, Fushimi-ku, Kyoto 612-8555, Japan; yoshida.kazuya.ut@mail.hosp.go.jp; Tel.: +81-75-641-9161; Fax: +81-75-643-4325

**Keywords:** botulinum toxin therapy, oromandibular dystonia, oral dyskinesia, bruxism, functional (psychogenic) movement disorder, palatal tremor

## Abstract

Various movement disorders, such as oromandibular dystonia, oral dyskinesia, bruxism, functional (psychogenic) movement disorder, and tremors, exist in the stomatognathic system. Most patients experiencing involuntary movements due to these disorders visit dentists or oral surgeons, who may be the first healthcare providers. However, differential diagnoses require neurological and dental knowledge. This study aimed to review scientific advances in botulinum toxin therapy for these conditions. The results indicated that botulinum toxin injection is effective and safe, with few side effects in most cases when properly administered by an experienced clinician. The diagnosis and treatment of movement disorders in the stomatognathic system require both neurological and dental or oral surgical knowledge and skills, and well-designed multicenter trials with a multidisciplinary team approach must be necessary to ensure accurate diagnosis and proper treatment.

## 1. Introduction

The stomatognathic system is an anatomic and functional unit composed of hard and soft tissues and has been studied in relation to occlusion and temporomandibular disorders [1,2]. The hard tissues include the bones forming the mandible and maxilla, dental arches, teeth, and temporomandibular joints, and the soft tissues include the masticatory, lingual, lip, cheek, and lower facial muscles, as well as the nervous and vascular supplies [3]. This system plays important roles in the various indispensable functions, such as chewing, swallowing, speaking, breathing, and in making facial expressions [3]. Dentists and oral surgeons specialize in the stomatognathic system and are the first healthcare professionals to see many patients with symptoms in that area. However, these professionals cannot always properly diagnose and treat involuntary movements in the stomatognathic system, as some cases require consultation with a neurologist, neurosurgeon, or psychiatrist [3,4].

Botulinum neurotoxin (BoNT) is the exotoxin of *Clostridium botulinum*, a Gram-positive, spore-forming bacterium. Kerner was the first to describe the symptoms of botulism in detail [5]. Van Ermengem isolated the microorganism *Bacillus botulinus* [6]. In 1979, Scott was the first to use BoNT therapeutically for strabismus via injection to the extraocular muscles [7]. The clinical applications of BoNT have since expanded to treat ophthalmic, neurological, gastrointestinal, urological, orthopedic, dermatological, dental, secretory, painful, cosmetic, and other diseases, and applications to the orofacial region have gained particular attention [8,9,10,11,12,13]. Target orofacial disorders include oromandibular dystonia (OMD) [14,15,16,17,18,19,20,21,22,23,24,25], hemifacial spasm [26,27], facial synkinesis [28], orolingual dyskinesia [29], functional (psychogenic) dystonia [30], trigeminal neuralgia [31,32,33,34,35,36,37], orofacial pain [38,39,40,41,42,43,44,45], temporomandibular disorder [46,47], temporomandibular joint dislocation [48,49], bruxism [50,51,52,53,54,55,56,57], palatal tremor [58,59], hypersalivation [60,61], spasmodic dysphonia [62,63], essential voice tremor [64], first bite syndrome [65,66], and Frey syndrome [67,68].

This study aimed to review the clinical problems, treatment challenges, and proposal for future studies of BoNT therapy for OMD, oral dyskinesia, bruxism, functional stomatognathic movement disorder (functional movement disorders in the stomatognathic system) [30], and palatal tremor from the perspective of an oral surgeon.

## 2. The Clinical Problem: History, Presentation, and Epidemiology

### 2.1. OMD

Dystonia is a hyperkinetic movement disorder characterized by sustained or intermittent muscle contractions that result in abnormal repetitive movements and/or postures [69]. Dystonia is categorized along two axes: (1) clinical characteristics, including the age at onset; body distribution (focal, multifocal, segmental, hemidystonia, and generalized); temporal pattern and associated features (additional movement disorders or neurological features), and (2) etiology: inherited, acquired, and secondary [69]. The term ‘dystonia’ was coined in 1911 by Oppenheim [70]. Before the concept of ‘dystonia’ was published, Romberg described various cases of masticatory muscle spasms due to a variety of pathogenesis in his textbook [71]. Among the cases, a few can be diagnosed as OMD, representing the first reports of this condition. The term ‘oromandibular dystonia’ was first introduced by Marsden in 1976 [72]. OMD is a focal type of dystonia characterized by contractions of the masticatory, lingual, pharyngeal, and/or muscles of the stomatognathic system [3,4,20,21,73,74,75,76,77,78,79,80].

#### 2.1.1. Presentation

OMD symptoms include masticatory disturbances, biting of the tongue or cheek membrane, limited mouth opening, pain or discomfort of the muscles, dysphagia, dysarthria, esthetic problems, upper airway obstructions [19], and temporomandibular joint dislocations [49]. Most of these symptoms can impair daily activities, social embarrassment, cosmetic disfigurement, inability to work, and unemployment, forming a significant impact on the overall quality of life of the patient [3,4,78,79,80]. Based on the site and direction of abnormal dystonic movements, OMD is classified into six subtypes: jaw closing, jaw opening, lingual, jaw deviation, jaw protrusion, and lip dystonia [3,4,78,80]. 

Italian movement disorder experts developed diagnostic recommendations for OMD, pointing to patterned and repetitive oromandibular movements/postures—either spontaneous or triggered by motor tasks [81]. If present, a sensory trick (geste antagoniste) confirmed the diagnosis of OMD [81]. In patients who did not manifest a sensory trick, the active exclusion of clinical features related to the condition mimicking dystonia was necessary for diagnosis [81]. OMD is usually diagnosed based on the characteristic clinical features of focal dystonia and electromyography (EMG) findings [3,73,78]. Patient clinical features include stereotypy, task specificity, sensory tricks, co-contraction, and morning benefit [3,4,78,82]. Patients with OMD exhibit different stereotypical patterns of muscle contraction according to subtype. Stereotypy was observed in 95.8% of 385 patients with OMD in one study [78]. OMD symptoms often appear as only task-specific, for example, while speaking or chewing, in its early phase. Notably, 69.9% of patients with OMD showed task specificity in one study [78]. A typical example of task-specific dystonia in the oral region is embouchure dystonia, which occurs only during the performance of wind instrument players [83]. Sensory tricks are physical movements or positions that can temporarily ameliorate the symptoms of dystonia [3,4,78,82]. Such sensory tricks were observed in 51.4% of patients with OMD in one study [78]. The symptoms of dystonia tend to be milder in the morning with large inter-individual variations in their duration; this phenomenon is called ‘morning benefit’ and was reported in 47.3% of patients with OMD [78]. Another symptom, co-contraction, refers to a loss of the reciprocal inhibition of muscular activities, causing involuntary simultaneous contractions of the agonist and antagonist muscles. Jaw closing muscle contraction during mouth opening can limit the maximal mouth opening, and the occurrence of such contractions while speaking or eating can hamper speech or mastication [3,4,78,82]. The overflow phenomenon involves the activation of muscles that are unnecessary for a task, interfering with normal movement [82]. Dystonic contracture of the masticatory muscles may expand to the orbicularis oris; orbicularis oculi; or other facial, neck, and shoulder muscles [82].

Dopamine receptor-blocking agents, including antipsychotics, tricyclic antidepressants, antiemetics, and other medications for gastrointestinal disorders, can induce tardive dystonia and represent common causes of acquired dystonia [84]. Some genetic disorders (DYT-*THAP1*, DYT-*TAF1*, DYT-*ATP1A3*, and DYT-*KMT2B*) are also characterized by OMD in the clinical spectrum [84]. 

#### 2.1.2. Epidemiology

The mean age of OMD onset is in the 50s [76,79,80]. Notably, women are approximately twice as frequently affected as men, indicating female predominance (2:1) [76,79,80]. OMD can occur in isolation; however, it may present together as segmental or generalized dystonia. For example, comorbid OMD and blepharospasm can form together as a segmental dystonia [85,86]. Common comorbidities of OMD include cervical dystonia, writer’s cramp, and spasmodic dysphonia [3,78,79,84]. The ratio of isolated OMD among all OMD cases differs considerably as reported by neurologists (focal, 39%; segmental, 43%; generalized, 10%) [79] and an oral surgeon (focal, 90.8%; segmental, 10.4%; multifocal, 6.3%) [80]. This discrepancy occurred because neurologists assessed several cases of OMD associated with neurological diseases, whereas the oral surgeon was able to identify numerous mild cases [80].

In a meta-analysis, Steeves et al. [87] estimated the prevalence of primary dystonia to be 16.43 per 100,000 persons, of cervical dystonia to be 4.98, of blepharospasm to be 4.24, and of OMD to be 0.52. The estimated prevalence of OMD varies from 0.1 to 6.9 per 100,000 populations [87,88,89,90,91,92,93,94,95,96]. The prevalence of OMD has been postulated to be considerably higher than previously estimated [3,78]. A recent study [80] reported the crude prevalence of OMD to be 9.8 per 100,000 persons (idiopathic dystonia, 5.7; tardive dystonia, 3.4), with an incidence of 2.0 per 100,000 person/years (idiopathic dystonia, 1.2; tardive dystonia, 0.68). The prevalence was 13.0 and 6.3 in 100,000 persons for women and men, respectively [80]. The study postulated that OMD may have an equal or even higher prevalence than cervical dystonia or blepharospasm [80]. Approximately 70% of patients with OMD visited dentists, and 60% of patients saw oral surgeons in one study [97]. However, approximately 90% of the patients had not been diagnosed correctly [97]. A vast majority of the patients had been diagnosed with temporomandibular disorders, bruxism, or psychiatric diseases [3,4,78,97]. Several patients continued the treatment of temporomandibular disorders or bruxism without effects, eventually discontinuing treatment and abandoning further consultations [3,4,78,97]. OMD is considered a rare disorder; however, in reality, the cases are simply incorrectly diagnosed.

### 2.2. Oral Dyskinesia

Dyskinesia is a general term for involuntary movements in which chorea, akathisia, tremor, ballism, athetosis, tic, and myoclonus occur in one or more combinations [98,99]. Historically, ‘dyskinesia’ referred to oro-buccal-lingual dyskinesia [98,99]. Subsequently, several involuntary movements were associated with this condition, and ‘dyskinesia’ became the general term for various movement disorders. This review focuses on oral dyskinesia. In the 1950s, neuroleptic medications were introduced, revolutionizing the treatment of psychiatric diseases, such as schizophrenia. Oral dyskinesias are often related to such drugs, particularly dopamine receptor-blocking agents, a category of medications that includes first- and second-generation antipsychotics [98,99]. This type of dyskinesia is called tardive dyskinesia and includes a group of delayed-onset iatrogenic movement disorders [98,99,100]. The term ‘tardive dyskinesia’ was first described in 1964 [101].

#### 2.2.1. Presentation

Oral dyskinesias are characterized by repetitive, involuntary, uncontrollable movements, such as lip pursing, pouting, smacking, and sucking, facial grimacing, tongue licking, writhing, protrusion and laterotrusion, and chewing-like motion [98,99]. Complications related to oral dyskinesia include mucosal traumatic lesions resulting from friction or biting injuries; inability to wear dentures; difficulties in speech, chewing, and swallowing; and social embarrassment [100]. In severe cases, repeated dyskinetic movements of the lower lip can result in the penetration of the skin [82]. Levodopa remains the primary medicine for controlling motor symptoms in Parkinson’s disease; however, complications cause dyskinesia [98,99,100,102]. Hyperkinetic movements are triggered by the ON phase and occur during the OFF phase or in both phases [102].

Tardive dyskinesia develops during exposure to drugs, such as dopamine receptor-blocking agents, including antipsychotics, tricyclic antidepressants, antiemetics, and other medications administered for gastrointestinal disorders, for at least three months (or one month in patients aged ≥60 years) or within four weeks of withdrawal from an oral medication (or within eight weeks of withdrawal from a depot medication) [103]. The risk of tardive dyskinesia is reported to be three to six times greater in elderly patients than in younger populations [104,105]. Patients with drug-induced dyskinesia tend to show severe lingual dyskinesia and grimacing, with more frequent extensions to limb or trunk musculature [99]. The pathophysiology of tardive dyskinesia remains poorly understood; however, striatal dopamine D2 receptor supersensitivity has been the widely-accepted explanation [98,99]. First-generation neuroleptics with high dopamine D2 receptor occupancy are thought to carry a higher risk of tardive dyskinesia than second-generation medications with low D2 receptor occupancy, such as clozapine and quetiapine [98,99]. 

Oral dyskinesia may occur in relation to neurodegenerative disorders (Huntington’s disease and neuroacanthocytosis) or neuropsychiatric conditions (chronic schizophrenia, Rett syndrome, and dementia) [99]. Furthermore, it may be induced spontaneously and peripherally (edentulousness or ill-fitting prosthesis) [99]. 

#### 2.2.2. Epidemiology

Dyskinesias were noted in nearly one-third of patients with schizophrenia prior to the introduction of antipsychotic medications [106]. The reviews showed that the prevalence of dyskinesia in patients with schizophrenia who had not been treated with antipsychotic drugs was 4.2% [107], 5% [108], or 12% [109]. The estimated rate of tardive dyskinesia in patients with schizophrenia who had been treated with antipsychotic medications was 15–20% [107,110] or 20–50%, with the prevalence increasing with advanced age [103]. The largest review conducted, involving 34,555 patients, reported an average prevalence of 20% [108].

Edentulousness has been thought to be a common cause of oral dyskinesia [99,111]. One study compared 75 edentulous participants (mean age: 62 years) seen in a dental clinic to 75 age-matched controls with natural teeth [111]. Overall, 12 (16%) of the edentulous participants showed oral dyskinesia, 9 exhibited mild dyskinesia, and 3 showed marked stereotypic dyskinesia. Six (50%) of the edentulous participants with oral dyskinesia had no dentures, and one participant wore ill-fitting dentures. 

In one study, 38 (3.7%) out of 1018 participants aged ≥60 years attending a daycare center were found to have oral dyskinesia without apparent cause [112]. The over-all prevalence was estimated to be 3.7% (4.1% for women and 2.9% for men) [112]. Participants with dyskinesia due to undefined causes reported more frequent ill-fitting prosthetics, oral pain, and a lower rate of perception of good oral health than those without oral dyskinesia [112]. In a survey of 352 edentulous elderly individuals, 7% displayed oral stereotypes, 4% reported tardive dyskinesia, and 2% exhibited antidepressant drug-related dyskinesia; furthermore, ill-fitting dentures were suggested as a possible triggering factor for the majority of patients [112]. Spontaneous oral dyskinesia is thought to be comparatively infrequent in the elderly population compared to rates in populations receiving antipsychotic drugs [99,107].

The prevalence of dyskinesia in untreated patients with chronic schizophrenia was estimated to be 12% in one review [108]. In another study [113], never-medicated patients with chronic schizophrenia were reexamined at 18 months, and a prevalence of dyskinesia 57% was reported. In other studies, prevalence ranged from 1.5% [114] to 31.7% [115]. The prevalence estimates of spontaneous orofacial dyskinesia in healthy elderly individuals show significant variations depending on the individuals evaluated, as with dystonia [87,88,89,90,91,92,93,94,95,96]. 

### 2.3. Bruxism

The term bruxism is derived from the Greek word ‘brychei’, which means to grind or gnash the teeth [116]. Bruxism is classified into sleep and awake bruxism, which are defined as masticatory muscle activities that occur during sleep and wakefulness, respectively [117]. Sleep bruxism is characterized as rhythmic or nonrhythmic, and awake bruxism is characterized by repetitive or sustained tooth contact and/or by bracing or thrusting of the mandible [117].

#### 2.3.1. Presentation

Although the etiology of bruxism has not yet been fully elucidated, a multifactorial etiology was postulated, including biological; psychological (anxiety and stress); and exogenous (consumption of some drugs, caffeine, tobacco, and/or alcohol) causes [117,118,119]. Bruxism has been recognized to be related to tooth wear or destruction; failure of dental prostheses or implants; pain in the teeth, masticatory muscles, or temporomandibular joints; temporomandibular disorders; masseter muscle hypertrophy; or tension headache [117,118].

#### 2.3.2. Epidemiology

Bruxism is a very common condition, observed to some degree during the lifetime of approximately 85–90% of the general population [120]. In many of these cases, the symptoms are temporary, the patients themselves are unaware of the symptoms, and active treatment is not required [117,118]. In earlier studies, the estimated prevalence of sleep bruxism awareness was based on reports by parents or sleep partners [121]. Recently, polysomnography with audio–video recording or ambulatory EMG devices have been used for a definite diagnosis [121]. Thymi et al. [121] suggested that, in future studies using ambulatory EMG instruments, the focus may require to shift to the concept of scoring the whole spectrum of masticatory muscle activity. 

A prevalence of 5.0% was found for awake bruxism and 16.5% for sleep bruxism [122]. A recent umbrella review reported that the prevalence of awake bruxism was 22–30% and that of sleep bruxism was 1–15%; sleep bruxism among children and adolescents was 3–49% [118]. Gender differences with respect to sleep bruxism are not obvious, as most studies reported an equal prevalence in men and women [123].

### 2.4. Functional Movement Disorder

Functional movement disorders are part of a spectrum of functional neurological disorders and are among the most common causes of neurological disability [124]. Recently, the term ‘functional’ has been used more frequently than ‘psychogenic’ [125]. These disorders are thought to be caused by a complex interplay of biopsychosocial vulnerabilities under exposure to psychosocial and/or physical triggers [126]. Charcot [127] first described the functional facial movement disorder as ‘l’hémispasme glosso-labié hystérique’. Although functional movement disorders often have characteristic clinical features in the orofacial region, their diagnosis is challenging [128]. Diagnoses of such conditions should rely not on the exclusion of organic diseases or the presence of psychological symptoms but on the observation of specific clinical features [126,129,130,131]. The misdiagnosis of OMD as a temporomandibular disorder or a psychogenic disease is frequent. Nevertheless, the situation in functional movement disorders in the stomatognathic system is more serious [30]. Thus, functional movement disorders in this region are under-recognized [30]. 

#### 2.4.1. Presentation

Despite the absence of specific electrophysiologic tests or a gold standard for diagnosis, functional movement disorders are diagnosed with clinically definite certainty based on the available criteria, regardless of any psychiatric symptoms [130,131]. Functional movement disorders often have characteristic clinical features in the orofacial region, including tonic muscular spasms, involving the lip, eyelids, perinasal region, and forehead [127,132,133,134]. The most common phenotype is tonic mandibular deviation accompanying ipsilateral downward and lateral lip pulling, seen in 84.3% of patients with facial functional movement disorders involving the craniofacial region [128]. Uni- or bilateral orbicularis oculi and platysma contraction are often associated [128].

In a previous study [30], 10-item inclusion criteria were formulated based on previously reported criteria for functional movement disorders [126,129,130,131] or clinical features in facial functional dystonia [128,132,133,134] to comprehensively assess 58 patients with functional movement disorders in the stomatognathic system. The criteria included 10 symptoms, and the prevalence of each was as follows: rapid onset, 74.1%; static course, 60.3%; paroxysmal symptoms, 86.2%; spreading to multiple sites, 89.7%; spontaneous remission, 58.6%; inconsistent symptoms, 93.1%; distractibility, 67.2%; incongruous symptoms, 91.4%; lack of sensory tricks, 81.0%; and suggestibility, 63.8% [30]. A functional dystonia phenotype (unilateral lower lip pulling and jaw deviation) was observed in 44.8% of the patients. The characteristic and distinguishable features of functional stomatognathic movement disorders included rapidly repeating lateral or tapping mandibular and tongue movements (27.6%), which considerably fluctuated with respect to the speed and direction [30].

The most prevalent complaint was pain (50%) in one study [30], while another study reported that 24.6% of patients complained of painful spasms [128]. Dysarthria (27.6%) and masticatory disturbances (15.5%) were also more common in this study than in previous reports [128,133]. This likely occurred because the study exclusively evaluated patients with symptoms in the stomatognathic region [30]. Depression was observed in 38% of patients with functional movement disorders in a previous study [128] and in 39.7% of patients in another study [30]. Frequent precipitating events included dental treatment (44.8%) and physical trauma (12.1%) [30].

#### 2.4.2. Epidemiology

Tremor, dystonia, myoclonus, and gait disturbance are the most prevalent presentations of functional movement disorders [126,129,130,131]. Such symptoms were seen in 5–20% of patients in a movement disorder clinic, functional dystonia being one of the most common [135]. Functional neurological disorders have an estimated prevalence of 50 per 100,000 population based on a community registry [124]. Involvement of the face represented 16.3% of all functional movement disorders in a large study [128]. Functional movement disorders of the orofacial region are prevalent in women (91.8% [128] and 72.4% [30]). The mean age at onset was 37 years [128] and 46.2 years [30]. 

Only one study has described functional movement disorders in the stomatognathic system [30]. In this work, 58 patients were diagnosed with functional stomatognathic movement disorders out of 1720 patients with complaints of involuntary movements or contracture of the masticatory, lingual, and/or lower facial muscles [30]. Therefore, the prevalence of this condition may be very low.

### 2.5. Palatal Tremor

Palatal tremors, which cause clicking tinnitus, were first documented by Boeck [136]. Two types of palatal tremor (also called palatal myoclonus) are described in the literature: essential and symptomatic palatal tremor [137,138]. Essential palatal tremor reveals no underlying structural pathology, whereas symptomatic palatal tremor occurs due to any lesion within the dentato–rubral–olivary pathway [139,140]. Essential palatal tremor occurs due to contraction of the tensor veli palatini muscle, supplied by the fifth cranial nerve, and symptomatic palatal tremor is due to the contraction of the levator veli palatini muscle, supplied by the ninth and tenth cranial nerves [141]. This review focuses on essential palatal tremors.

#### 2.5.1. Presentation

Palatal tremors are involuntary contractions of the soft palate often accompanied by a clicking sound [59,142]. Palatal tremor is usually attributed to a dysfunction or lesion in the Guillain-Mollaret triangle; however, many other causes exist, including functional (psychogenic) factors [143]. Symptoms include clicking tinnitus, nonaudible awareness of palatal movements, and rhinolalia [59]. The most troublesome symptom for the patient is synchronous audible clicking tinnitus (ear clicking) accompanying the abnormal movements [59]. An essential palatal tremor is much more likely to present with a clicking tinnitus than a symptomatic tremor [59]. Clicking tinnitus is produced by contractions of the tensor veli palatini muscle, which opens the Eustachian tube, causing a sudden drop in surface tension within the tube [138,140]. 

#### 2.5.2. Epidemiology

Zadikoff et al. [140] reviewed the existing literature on essential palatal tremor and found a male:female ratio of 1:1. Sinclair et al. [59] reported that patients were more commonly female (60% vs. 40%). Deuschl et al. [138] reported the persistence of essential palatal tremor during sleep in 50% of patients. The mean age at symptom onset was 35.6 years, including childhood onset. The essential palatal tremor began after a viral respiratory infection in 40% of patients [59]. The frequency of essential palatal tremor is not only highly variable among different patients; however, it can also vary within a single individual [139].

## 3. Treatment Challenges and Pitfalls

### 3.1. OMD

OMD can be part of the clinical spectrum of various neurological diseases, including Parkinson’s disease, pantothenate kinase associated neurodegeneration, Wilson’s disease, chorea-acanthocytosis, Lesch Nyhan syndrome, and Leigh syndrome, or of ischemic or hemorrhagic stroke, tumors, infarction, and brain injury [84]. If such diseases have already been diagnosed and treated, OMD must be addressed simultaneously by the attending physicians [3,4]. However, if not diagnosed yet, the patient should be referred to specialists [3,4]. Likewise, collaboration with a psychiatrist is required for the treatment of tardive dystonia. 

The treatment of OMD must be multimodal and highly individualized, and the methods include pharmacological [4,82,84], BoNT [14,15,16,17,18,19,20,21,22,23,24,25,75,144,145,146,147,148,149,150,151,152,153,154,155], muscle afferent block [73,156,157], occlusal splint [158,159], and surgical therapies (coronoidotomy) [160,161,162]. Due to the number of muscle spindles, muscle afferent block therapy is more effective for jaw closing muscles than for jaw opening muscles [73,156]. A sensory trick splint is particularly successful in patients with muscle hyperactivity of the jaw closing muscles. In one study, 83.7% of the responders with splints presented with jaw closing dystonia [158]. Patients who showed improvement with the use of splints and continued to wear them for at least three months were defined as responders [158], while patients who showed little or no effect and/or were unable to insert splints were defined as non-responders [158]. Intraoral sensory tricks were significantly more common in responders (60.2%) than in non-responders (13.3%) [158]. Coronoidotomy is only indicated for severe jaw closing dystonia [160,161]. However, one-third of the operated patients required additional BoNT injections into the masseter and/or medial pterygoid muscles [161]. This review will focus on BoNT therapy. 

Chemodenervation with BoNT is considered the first line of treatment for OMD. Currently, four FDA-approved and commercially available BoNT formulations. The three types of botulinum toxin type A available are onabotulinumtoxinA (Botox; Allergan, Irvine, CA, USA), abobotulinumtoxinA (Dysport; Ipsen-Pharma, Berkshire, UK), and incobotulinumtoxinA (Xeomin; Merz Pharma, Frankfurt am Main, Germany); rimabotulinumtoxinB (Myobloc in the USA; Supernus Pharmaceuticals, Inc, Rockville, MD; and Neurobloc in Europe, Sloan Pharma, Baar, Switzerland) is a botulinum toxin type B preparation [13,163,164]. The following ratios are often used in clinical practice: onabotulinumtoxinA:incobotulinumtoxinA = 1:1; onabotulinumtoxinA: abobotulinumtoxinA = 1:2.5, and onabotulinumtoxinA:rimabotulinumtoxinB = 1:50 [13,164]. Although BoNT is frequently used in major countries for treatment, it has not been officially approved for OMD. 

For the first injection, the dose of BoNT must be low because effects vary between individuals [4]. In the subsequent injections, the dose should be adjusted individually corresponding to the effects to reduce the risk of side effects or antibody development and minimize the cost. A risk of developing neutralizing antibodies exists with the long-term use of BoNT [13,165]. The factors that increase the risk of developing resistance to include a high protein load in some formulations, large individual and cumulative doses, and short intervals [165,166,167,168]. Since the dose for OMD is relatively small, the risk of antibody development is low; however, a case was reported of antibody development after a large amount of BoNT was injected intensively [15]. The contraindications of BoNT include systemic neuromuscular junction disorders (myasthenia gravis, Lambert-Eaton syndrome, and amyotrophic lateral sclerosis); current or possible pregnancy; and lactation.

To predict the efficacy of injected BoNT, 3–5 mL of 0.5% lidocaine was injected into the hyperactive muscles of patients in one study, and involuntary movements were carefully observed [45]. If the patients showed changes at all, BoNT was considered not likely to be effective. If the patients showed an improvement of symptoms under the effects of the local anesthetic, subsequent appointments for BoNT injections were recommended [45]. 

A complete understanding of the local anatomy of the stomatognathic system is a prerequisite for target muscle selection and safe injection without complications [4,20,25]. BoNT is reconstituted with normal saline to reach a final concentration of 2.5–5 units/0.1 mL. Differences in the concentration may affect diffusion; however, no difference in effect was seen in a study for blepharospasm [169]. A disposable hypodermic needle electrode (37 mm × 25 G, 50 mm × 25 G) is used for injection into the target muscles, and accurate placement of the electrode is verified by evaluating EMG activity [4,20,25]. Subsequently, after aspiration, the necessary units of botulinum toxin are injected into the target muscles.

Sonography is important for identifying target muscles and preventing damage to other tissues and is often used for cervical dystonia [170,171]. This method is also very useful for OMD and is recommended for use in combination with other methods such as EMG.

The maximum bite force is measured bilaterally on the molars three times using an occlusal force meter [4,45,172]. The muscles and doses of BoNT are then individually determined for each patient based on their symptoms and occlusal force [4,45,78,172]. The injection is continued until the patient is satisfied with the effect [4,45]. The injection interval is three to six months, depending on patient symptoms [4,45,172]. 

#### 3.1.1. Jaw Closing Dystonia

Jaw closing dystonia was observed in 59.5% of patients with OMD [78]. In severe cases, the patients cannot open the mouth at all because of the involuntary contraction of the bilateral temporalis and masseter muscles [160,161]. The recommended target muscles and doses of BoNT for jaw closing dystonia [4,45,173] are summarized in Table 1.

Masticatory disturbance may be a side effect of an excessive loss of occlusal force. The maximum occlusal force should be measured before and after treatment with BoNT to prevent an excessive loss of chewing force [4,172].

##### Masseter Muscle

The main target muscle of BoNT treatment for jaw closing dystonia is the masseter muscle, which comprises a superficial and a deep part [4,173]. Posteriorly, the muscle is covered by the parotid gland, and care must be taken not to damage it with a needle [4,173]. An ultrasound-guided injection makes it possible to distinguish the masseter muscle from the parotid gland [4,173]. BoNT is injected 10–15 mm into the most prominent region during jaw clenching, with 10–50 units (depending on the condition) at three points (Figure 1) [4,173].

##### Temporalis Muscle

After careful examination including EMG, occlusal force and palpation, BoNT is often injected into the temporalis muscle at the same time. This muscle, particularly anterior fibers, elevates the mandible, and pulls it back with posterior fibers. Notably, 10–50 units of BoNT are injected 10–15 mm deep at three sites (Figure 1) [4,173].

##### Medial Pterygoid Muscle

Repeated injections into the masseter and temporalis muscles can cause tension on the medial pterygoid muscle due to the ‘whack-a-mole phenomenon’ [161,174]. This phenomenon can be explained as follows: When a dystonic muscle has been improved by BoNT treatment, one of the other muscles with the same function becomes more active [161,174]. Subsequently, when BoNT is injected into the latter muscle, another muscle with a similar function gradually becomes dystonic [161,174]. It is therefore necessary to inject BoNT into the medial (internal) pterygoid muscle. The medial pterygoid muscle is accessible via intraoral and extraoral approaches (Figure 2). With the intraoral method, the patient must gargle with a mouthwash solution; then, the needle should be positioned at an angle of 20° to the rear and upward and then to the side by 20° after palpation of the muscle in relation to the occlusal plane (Figure 2) [173]. The needle should be inserted up to a depth of 15–20 mm. BoNT is injected in 10–30 units. The correct placement of the needle in the target muscle should be monitored by EMG [161,173,174]. For the extraoral method, the patient’s head should be tilted towards the contralateral side. The needle is then inserted into the submandibular skin, 10 mm forward from the angle of the lower jaw, parallel to the inner side of the mandible, and at a depth of 15–20 mm (Figure 2) [161,173,174].

Patients who mainly exhibit grinding also often report severe tenderness in the lateral pterygoid muscle. The details of this muscle are described in Section 3.1.2.

#### 3.1.2. Jaw Opening, Deviation, and Protrusion Dystonia

The occurrence rates of jaw opening, jaw deviation, and jaw protrusion dystonia in patients with OMD are 12.7%, 5.5%, and 3.1%, respectively [78]. In jaw opening and protrusion dystonia, the bilateral lateral pterygoid muscles show involuntary hyperactivity [22,173,174]. In contrast, in jaw deviation dystonia, the muscle on the contralateral side of the deviation exhibits abnormal contractions [22,173,174]. In the most severe cases, the temporomandibular joint is dislocated [19,49]. The recommended target muscles and doses of BoNT for jaw opening, jaw deviation, and jaw protrusion dystonia [4,22,49,173,174] are summarized in Table 2.

##### Lateral Pterygoid Muscle

The lateral (external) pterygoid muscle is a two-headed muscle comprising a superior (upper) and an inferior (lower) head [22,173,174]. As the bilateral inferior heads contract, the condyle is pulled forward and slightly downward. If the muscle is only activated on one side, the inferior jaw rotates around a vertical axis that runs through the contralateral condyle, and it is pulled medially to the contralateral side [22,173,174]. The superior and inferior heads are activated alternately during chewing, such that the inferior head contracts during mouth opening while the superior head relaxes. Upon closing of the mouth, the actions are reversed. [22,173,174,175,176].

The maxillary artery arises behind the neck of the mandible and is initially embedded in the parotid gland [22,173,174]. Known anatomical differences exist between Japanese and Caucasian populations in the course of the maxillary artery in relation to the lateral pterygoid muscle [22,173,174]. In 92.7% of Japanese individuals, the maxillary artery runs laterally to the lower head [22,173,174]. However, in Caucasians, the artery more often runs medially to the muscle (38%) [22,173,174]. Injury to the maxillary artery during needle insertion can result in arterial bleeding, swelling, and bruising. Both intraoral and extraoral methods are available for BoNT injection; when injecting into the lateral pterygoid muscle, medical practitioners must be aware of these anatomical differences (Figure 3) [22,173,174].

The extraoral injection is easy to administer; however, BoNT can only be injected in a limited area [22,173,174]. Further, a risk of mouth dryness exists [147]. In contrast, the intraoral injection can be applied directly into the region with muscle tension with EMG monitoring. With the intraoral method, the needle is inserted 20–30 mm into the inferior head of the mucobuccal fold next to the upper second molar [22,173,174]. The angle of needle insertion is 30° upwards against the occlusal surface and 20° inwards against the sagittal plane (Figure 3). BoNT is injected in 10–50 units, depending on the condition. With the extraoral method, after palpating the infratemporal fossa, the needle can be inserted perpendicular to the skin 20–30 mm deep through the notch in the mandible (Figure 3) [22,173,174].

Computer-aided design and manufacturing process was used to develop a needle guide to reliably administer BoNT into the inferior head of the lateral pterygoid muscle [22,173,174]. Computed tomography and scan data of the upper jaw model are transferred to a computer, and the two most suitable points on the lower head are determined [22]. The needle guide is subsequently mounted in the oral cavity, and the needle is inserted to the planned depth. The experiments showed that the needle was easily inserted without any complications in all the procedures [22].

##### Digastric Muscle

The digastric muscle comprises two muscle bellies (anterior and posterior) connected by an intermediate tendon [4]. The anterior belly of the digastric muscle acts in the early phase of opening the mouth by lowering the chin [175,176]. Injection into the anterior belly can cause dysphagia and should be carefully monitored. BoNT is injected in 5–10 units [4]. Patients with OMD or bruxism have often reported tenderness in the posterior belly of the digastric muscle. If the patients have tenderness in the muscle, the muscle can be easily palpable. Since blood vessels and nerves run nearby, approximately 2.5–5 units of BoNT should be carefully injected under EMG guidance [4].

##### Temporalis Muscle and Platysma

If the effect of BoNT injections not satisfactory for patients with jaw deviation dystonia, BoNT (10–20 units) should be injected into the posterior fiber of the ipsilateral temporalis muscle. In some patients with jaw opening dystonia, the platysma may become hyperactive and require injection (10–20 units). 

#### 3.1.3. Lingual Dystonia

Lingual (tongue) dystonia is characterized by involuntary, often task-specific contractions of the tongue muscle [177]. Lingual dystonia was found in 25.5% of patients with OMD [78]. Lingual dystonia is divided into four subtypes: protrusion, retraction, laterotrusion, and curling [25]. 

The tongue comprises four extrinsic (genioglossus, hyoglossus, styloglossus, and palatoglossus muscles) and four intrinsic (superior longitudinal, inferior longitudinal, transverse, and vertical muscles) muscles [25,178]. The extrinsic muscles control tongue position, while the intrinsic muscles control movement [25,178]. Although serious complications of BoNT injection, including life-threatening dysphagia, aspiration pneumonia, and breathing difficulties, were reported in previous studies [179,180], the therapy has been recognized as a promising option in recent years [25,181,182]. 

The subjective improvement after the injection of BoNT was 77.6% in 136 patients with lingual dystonia [25]. The greatest improvement was seen for curling type dystonia (81.9%), whereas the retraction type exhibited the lowest improvement rate (67.9%) [25]. Mild dysphagia occurred in 12.5% of the patients; however, this resolved spontaneously within a few days to two weeks. No serious side effects were observed [25]. Lingual dystonia is often accompanied by jaw opening dystonia. For such patients, injection into the lateral pterygoid muscle is necessary [4]. The recommended target muscles and doses of BoNT for lingual dystonia [4,25,173] are summarized in Table 3.

##### Genioglossus and Other Tongue Muscles

The genioglossus muscle is the dominant muscle for tongue protrusion [178]. BoNT injection into the lingual muscle occasionally causes life-threatening complications, such as serious dysphagia [179], aspiration pneumonia [179], and swallowing or breathing difficulties [180]. Investigators have reported various methods for BoNT injection, primarily the submandibular approach [25,179,180,181,182,183,184,185], with intraoral approaches also reported [186]. In the submandibular method, injection is performed in one or two sites bilaterally. 

A more detailed injection method for the four subtypes of lingual dystonia has been described, depending on the direction of the tongue deviation [25,173]. The dose of BoNT therapy should be started at 10–20 units and gradually increase to 40–50 units, corresponding to patient symptoms [25,173]. The appropriate doses of BoNT (15–60 units) is then determined based on EMG examination and patient symptoms (Table 3). 

For protrusion type dystonia, approximately 50–100% of the total doses are injected into the bilateral genioglossus percutaneously through the submandibular region. The insertion points are defined as two sites 25–30 mm posterior from the midline of the body of the mandible and 15–20 mm apart from each other (Figure 4A) [25]. If tongue protrusion occurs while simultaneously curling up or down, the remaining doses are injected into the superior longitudinal muscle (5 mm depth; injection to counteract tongue protrusion) or into the inferior longitudinal muscle (10–15 mm depth; injection to counteract curling) [25]. If laterotrusion occurs with protrusion, BoNT is administered into the superior and inferior longitudinal muscles on the deviated side (Figure 4B) [25]. If the tongue shows flattening or narrowing, the remaining BoNT is injected into the bilateral vertical muscle (10 mm in depth; to counteract protrusion) and the bilateral transverse muscle (10 mm in depth; to counteract curling) [25]. 

For retraction type dystonia, target muscles are identified after careful EMG examination. These include a wide range of tongue muscles that may undergo contraction, such as the genioglossus, intrinsic muscles, geniohyoid, and hyoglossus [25]. Appropriate doses of BoNT are in the range of 15–50 units (Table 3). 

For laterotrusion type dystonia, the appropriate dose of BoNT is between 10 and 40 units (Table 3). BoNT is injected into the superior (5 mm in depth) and inferior (10–15 mm in depth) longitudinal muscles on the deviated side (Figure 4B) [25]. The inferior longitudinal muscle is more accessible from the inferior aspect of the tongue (5 mm in depth) than from the dorsum. If the genioglossus on the opposite side exhibits EMG activity, an additional injection is administered [25]. 

For curling type dystonia, the appropriate dose of BoNT is between 10 and 40 units (Table 3). The toxin is injected bilaterally at two or three sites from the dorsum of the tongue, approximately 5 mm in depth, into the superior longitudinal muscle (Figure 4B) [25]. BoNT is injected into the superior longitudinal muscle near the apex if curling up of the apex occurs (Figure 4B).

#### 3.1.4. Lip Dystonia

Lip dystonia was observed in 3.6% of the reported OMD cases [78]. The orbicularis oris muscle is a sphincter that surrounds the oral orifice. According to the symptoms, BoNT should be injected into the orbicularis oris, risorius, depressor anguli oris, depressor labii inferioris, mentalis, and platysma. Since the fibers of the orbicularis oris muscle close to the orifice are responsible for pursing the lips, and the fibers distant from the orifice press the lips against the teeth [149], the dose and site should be adjusted according to the symptoms [4]. The dose of BoNT should be low (2.5–5 units) because of risk of the labial incompetence and asymmetric smile.

Lip dystonia, especially unilateral traction, can occur with functional involuntary movements [30]. Functional movement disorders often show distinguishable clinical features in the orofacial area. Treatment with BoNT is effective in some cases, but complete treatment often requires psychotherapy or physiotherapy [30].

#### 3.1.5. Pitfalls

The largest clinical pitfall of OMD treatment seems to be a misdiagnosis. Most patients with OMD see a dentist or oral surgeon and are diagnosed with bruxism, temporomandibular disorders, and/or psychiatric disorders [78,97]. OMD is considered a blind spot between medical science and dentistry [78]. Neurologists may diagnose temporomandibular disorder as OMD, (*i.e.*, bruxism as jaw closing dystonia and anterior disc displacement without reduction as jaw deviation dystonia) [78]. The knowledge and experience of both neurology and dental medicine are needed for proper diagnosis of OMD from temporomandibular disorders, but these two perspectives are nearly impossible to obtain simultaneously. Therefore, a simple diagnostic tool was developed to enable primary care physicians, neurologists, dentists, and oral surgeons to differentiate OMD from temporomandibular disorder and to initiate appropriate treatments rapidly [78].

A subsequent pitfall of BoNT treatment is the method of injection into the masticatory muscles. Dressler et al. [155] described in their consensus guidelines, “The Mm. pterygoidei can easily be injected through the incisura mandibulae. EMG requiring thick combination needles seems unnecessary as dystonic involvement is usually affecting both, the lateral and the medial pterygoid muscles.” Nevertheless, it is risky to inject BoNT into the lateral and medial pterygoid muscles without EMG guidance [22,173,174]. The lateral pterygoid muscle should be injected in jaw opening dystonia [22,173,174]. In contrast, the medial pterygoid muscle should be injected for patients with jaw closing dystonia [21,74,147,173]. If BoNT is erroneously injected into the lateral pterygoid muscle for jaw closing dystonia, or into the medial pterygoid muscle for jaw opening dystonia, adverse effects are expected. Since the needle is inserted at the notch by percutaneous lateral pterygoid muscle injection, penetrating the parotid gland, mouth dryness may occur due to the spread of BoNT [147]. Most OMD specialists recommend an intraoral approach [20,21,22,49,74,147]. 

In a systematic review [77] involving 387 patients with OMD treated with BoNT, 27.1% of the patients had side effects, most frequently dysphagia. Previously reported adverse effects of BoNT/A include temporary regional weakness, tenderness over the injection sites, minor discomfort during chewing, asymmetric smile, loss of smile, lip numbness, muscle atrophy, paresthesia, difficulty in swallowing, mouth dryness, speech changes, nasal speech, headache, hematoma, nasal regurgitation, swelling, bruising, facial asymmetry, transient edema, itching, and pain at the injection area [15,16,18,21,139,147,150,187]. Most of these effects were observed to be transient and spontaneously disappeared. Moreover, the majority of these side effects are thought to be related to the injection technique and avoided by accurate knowledge of the local anatomy of muscles, nerves, and other tissues and accurate injection procedures. The more accurately BoNT is administered into the target muscles, the more likely the improvement in patient symptoms, and the lower the risk of complications [22,25,45,49]. Empirical differences in injection techniques may be associated with adverse effects [45]. Selected dose should be as high as necessary but as low as possible.

Several authors have reported changes in the mandibular bone after BoNT injections [188,189]. In contrast, a retrospective study in adult women with masseter hypertrophy found no significant change in whole mandible volume or in cortical thickness of the mandibular ramus three months after BoNT therapy in the masseter muscles [190]. Changes in bone after BoNT therefore may not be an adverse effect but a normal physiological response related to the corrected occlusal force [45]. Injection with BoNT improves the excessive tension of the jaw elevator muscles and allows the hypertrophied muscles and bones to return to their original shape [45]. 

Since the symptoms of OMD can vary from patient to patient, it is difficult to objectively measure disease severity and changes after treatment [3]. In 2002, 44 patients with OMD were evaluated before and after muscle afferent block therapy using a simple clinical scoring system according to subscores for pain, mastication, speech, and discomfort [156]. In 2010, Merz et al. [191] developed and validated the Oromandibular Dystonia Questionnaire. Recently, a comprehensive measurement tool for OMD, the ‘Oromandibular Dystonia Rating Scale’ was developed and validated [3]. The scale can be useful for the comprehensive evaluation of severity, disability, psychosocial functioning, and impact on the quality of life, as well as therapeutic changes in patients with OMD [3]. 

### 3.2. Oral Dyskinesia

Preventing tardive dyskinesia during psychiatric drug administration is paramount, and the strict selection of patients to be treated with dopamine receptor-blocking agents is a prudent medical practice [98,99]. In patients receiving dopamine receptor-blocking agents, if discontinuation is possible, the drug may be delayed by slowly reducing the dose over several months [98]. Tardive dyskinesia symptoms are often alleviated but can persist indefinitely after drug withdrawal. The younger the age or the shorter the administration period, the higher the remission rate. However, a drug having a dopamine-depleting effect should be used if the medication cannot be discontinued. 

Switching from a first-generation to a second-generation antipsychotic with lower D2 affinity, such as clozapine or quetiapine, may be efficacious in reducing tardive dyskinesia symptoms [98,99,192]. Chemodenervation with BoNT injections into the masticatory or lingual muscles causing dyskinesia has also been successfully applied [29]. BoNT should be injected into jaw closing, jaw opening, or tongue muscles or into muscles around the mouth (orbicularis oris, risorius, zygomatic, and mentalis muscles), according to the methods described in Section 3.1.1, Section 3.1.2 and Section 3.1.3. If the symptoms are severe and all drug therapies are found to be ineffective, deep brain stimulation, such as globus pallidus stimulation, may be performed [98]. However, there is limited evidence for the use of globus pallidus interna deep brain stimulation.

In addition, anticholinergic drugs may be used to treat idiopathic oral dyskinesia and rabbit syndrome [99]. However, the symptoms of patients with oral dyskinesia often worsen with anticholinergics. Further, elderly patients require careful monitoring of symptoms, such as dementia and constipation caused by the anticholinergic effect of this drug [99].

If orolingual dyskinesia causes the tongue or lips to come into contact with the teeth and form ulcers [82], and other therapies are ineffective, a protective guard, such as a splint, is inserted as a dental symptomatic treatment, or occasionally, the causative tooth is extracted. Furthermore, chronic undesired peripheral sensory inputs in the stomatognathic system caused by ill-fitting dentures and edentulousness, are thought to induce dyskinesia [99]. In edentulous patients with oral dyskinesia inserting dentures, adequate adjustment or relining of the dentures has successfully improved the intensity of the symptoms [193,194].

#### Pitfalls

Neurological, psychiatric, and dental factors may contribute, to a variable extent, to oral dyskinesia [99]. A multidisciplinary evaluation with a neurologist, psychiatrist, and dentist or oral surgeon is recommended for patient management. 

### 3.3. Bruxism

Bruxism should not be recognized as a disorder in otherwise healthy individuals but rather as a behavior that can be a risk (and/or protective) factor for certain clinical consequences [117]. Treatments for bruxism include medication, occlusal splint, physical therapy, and cognitive behavioral therapy [118]. In recent years, BoNT therapy has been clinically applied [50,51,52,53,54,55,56,57]. Most patients with bruxism experience symptom relief via traditional methods, such as oral medications and splints. BoNT therapy should be considered in severe cases when other treatments are ineffective [172]. Prolonged, severe, and/or intense bruxism can cause the develops excessive tendonous tissue at the anterior margin of the masseter muscle, resulting in masticatory muscle tendon-aponeurosis hyperplasia and severe trismus [161,162]. In such cases, coronoidotomy is required [160,161,162].

The masseter muscle contributes to approximately 43% of the intrinsic strength of the jaw elevator muscles, to approximately 36% of the strength of the temporalis muscle, and approximately 21% of the strength of the medial pterygoid muscle [195]. Palpation, EMG, and occlusal force measurement are used to determine the target muscles [4,45,172]. In most cases, the masseter and temporalis muscles are injected first. In cases of medial pterygoid muscle tenderness, continuous injections into the masseter and temporalis muscles are administered, and if the effect diminishes, the medial pterygoid muscle is treated [4,172,173,174]. In cases with severe grinding, the lateral pterygoid muscle is often sensitive, and the lateral pterygoid muscle is also injected [4,172,173,174]. The occlusal force must be measured before and after each BoNT injection, as long-term repeated injections can result in the drop of occlusal force and masticatory disturbance [45,172].

In one study, 108 patients (33 men, 75 women) with involuntary contractions of the masseter, temporalis, and medial pterygoid muscles were treated for a total of 342 injections of BoNT [172]. The patients were refractory to oral medications, splints, myomonitor, and other treatments. After BoNT injection, the involuntary muscle tone and related myalgia were alleviated, and the maximum occlusal force between the maxillary and mandibular first molars was significantly reduced on the left (410.9 N→291.3 N) and the right (433.5 N→313.8 N) sides [172]. The average subjective improvement (self-evaluation where 0% no effect and 100% complete cure) was 82% [172]. Ahn et al. [196] studied seven patients with masseteric hypertrophy. Twenty-five units of BoNT was injected into each masseter muscle, and the differences in maximum occlusal force between the pre-injection and 2-, 4-, and 8-week post-injection values were statistically significant. However, no significant difference was seen between the values at baseline and after 12 weeks. Although the maximum occlusal force significantly decreased after injection of BoNT for the treatment of masseter hypertrophy, force gradually recovered by 12 weeks [196].

BoNT therapy is important in severe cases of bruxism, but a surgical procedure under general anesthesia should be considered in persistent, severe cases where mouth opening is completely impossible due to forceful clenching [160,161].

#### Pitfalls

A significant number of patients diagnosed with awake bruxism may exhibit jaw closing dystonia [3,78]. The treatment of bruxism in patients with jaw closing dystonia has a limited effect, and the differential diagnosis between awake bruxism and jaw closing dystonia is important [78]. Notably, several dental professionals are interested in and studying bruxism but not OMD. Therefore, raising awareness of OMD among dentists and oral surgeons, who are likely to see patients with OMD first, is critical.

Adverse reactions from BoNT injections are uncommon, localized, and dose-dependent [197]. A study by Ondo et al. [56] found a change in the smile of two participants. Shim et al. [54] reported three participants with masticatory difficulties. In a retrospective study of 2036 BoNT treatment sessions for masseter hypertrophy [198], the main complication was perceived muscular weakness and aching, which was observed in 30% of patients. The second-most common complication was bruising due to needle puncture of vessels in the soft tissue, occurring in 2.5% of the injections. All other complications occurred in less than 1% of cases and included headache, asymmetrical smile, limited mouth opening, and xerostomia [198]. Recent studies have reported an association between mandibular bone loss and the use of BoNT in the masticatory muscles [188,189]. However, as described above, the changes in the bone after BoNT injection are not considered an adverse effect but a normal physiological response related to the corrected masticatory force.

An asymmetrical smile may be related to the spread of BoNT into the risorius or zygomatic muscles. Xerostomia can occur due to the spread of BoNT into the parotid gland [147]. Special attention should be paid during injection into the masseter muscle and during extraoral injection to the lateral pterygoid muscle [4,147].

### 3.4. Functional Movement Disorder

The treatment of functional movement disorders should begin with explaining the diagnosis and ensuring patient understanding [130]. Treatments include antidepressants, psychological therapy in the form of psychodynamic psychotherapy or cognitive behavioral therapy, and transcranial magnetic stimulation [126,130,199]. Comorbid depression, anxiety, and pain may be treated pharmacologically. LaFave et al. [200] reported the most effective therapeutic options for functional movement disorders to be avoiding iatrogenic harm (58%) and educating patients about their diagnosis (53%), based on a recent survey of members of the International Parkinson and Movement Disorder Society.

In a study of 55 patients with functional facial movement disorder, spontaneous remission (a characteristics of functional movement disorder) was observed in 21% of the patients. Notably, 56% of the patients did not show any improvement; 20% exhibited worsened condition; and 20% showed improvements (BoNT injection in five, antidepressants in three, antiepileptics in two, and psychotherapy in one) [128]. In a study of 58 patients with functional stomatognathic movement disorders, symptomatic therapy according to the comprehensive treatment of OMD was selected depending on presenting symptoms included medication, muscle afferent block therapy, BoNT therapy, occlusal splint use, and myomonitor (transcutaneous electro–neural stimulation) [30]. Therapeutic effects were unsatisfactory in most cases in the study [30]. If therapies resulted in ineffective or unsatisfactory responses, patients were referred to psychiatrists or acupuncturists [30].

When patients showed apparent muscle hyperactivity, BoNT therapy improved symptoms considerably [30]. For typical cases of jaw or tongue deviation, BoNT should be administered to the lateral pterygoid or tongue muscles. Additional target muscles include the platysma, orbicularis oris, risorius, mentalis, zygomaticus, and depressor anguli oris muscles [30]. Occlusal splints are occasionally effective for patients with a sensory trick in the oral cavity [158]. Positive effects including immediate improvement just after muscle afferent block therapy or insertion of the splint may be related to suggestibility or the placebo effect, which are typical clinical features in functional movement disorders [30]. 

#### Pitfalls

In one study, only 5.2% of patients with functional stomatognathic movement disorders were suspected to have functional movement disorders despite diagnoses by movement disorder specialists [30]. The patients in the study visited 6.2 hospitals over 4.1 years [30]. The entity is frequently misdiagnosed as other psychiatric, neurological diseases. 

A multidisciplinary team approach involving a neurologist, psychiatrist, neurosurgeon, physiotherapist, psychotherapist, and oral surgeon is preferable for the diagnosis of and individual therapies for functional stomatognathic movement disorders [30]. However, very few psychiatrists or psychotherapists are willing to treat patients with such conditions [201]. The survey to members of the International Parkinson and Movement Disorder Society [200] suggested frequent treatment barriers, including a lack of physician knowledge and training (32%), lack of treatment guidelines (39%), limited availability of referral services (48%), and cultural beliefs about psychological illnesses (50%). A lack of access to both mental healthcare providers and rehabilitation specialists remains an important limitation for treatment [202].

### 3.5. Palatal Tremor

Management options for essential palatal tremors include medication (clonazepam, lamotrigine, sodium valproate, and flunarizine) [141]; psychotherapy [203]; and surgical procedures; however, the tinnitus and palatal movements are often refractory [59]. Several case reports and series suggest BoNT is effective in the treatment of palatal tremors [59,204]. One retrospective study included 15 patients with essential palatal tremors, and 2.5 or 5 units of BoNT caused amelioration of symptoms in 85.7% of the cases [59]. BoNT injection into the tensor veli palatini muscle is particularly useful for patients with tinnitus, whereas BoNT injection into the levator veli palatini muscle is efficacious for patient-perceived palatal motion [59].

#### Pitfalls

The adverse effects of BoNT therapy include voice change, nasopharyngeal regurgitation [58], Eustachian tube dysfunction, and velopharyngeal inadequacy [205]. A well-experienced and knowledgeable clinician should inject BoNT and should be prepared to handle any complications.

## 4. Reported Trials—Evidence-Based Medicine

### 4.1. OMD

In a double-blind, placebo-controlled study of BoNT treatment for cranial–cervical dystonia in 10 patients with oromandibular–cervical dystonia, only 37.5% of patients reported improved symptoms [14]. However, the low number of participants and combined phenotype of cervical dystonia and OMD in this study limit the conclusions [13]. Tan and Jankovic [18] studied 162 patients with OMD for approximately 10 years. BoNT was administered in the submentalis muscle complex of patients with jaw opening dystonia and in the masseter muscles patients with jaw clenching with or without bruxism. A total of 31.5% of patients experienced adverse effects (dysphagia, 10.2%; dysarthria, 0.9%) [18]. A retrospective chart review included 59 patients with OMD (jaw closing, 47.5%; jaw opening, 35.6%; and jaw deviation, 16.9%) [21]. Bakke et al. [20] reported functional and clinical characteristics in 21 patients with OMD. Fourteen patients received BoNT for OMD for 8–10 years, and 9 out of 14 patients continued with BoNT therapy [154]. Intraoral BoNT injection into the lateral pterygoid muscle in six of the eight patients with OMD led to significant symptom improvement, and only one patient experienced nasal speech [150].

More recently, a pilot single-blind study was published evaluating BoNT dosing and efficacy in 18 patients with OMD (jaw opening, 9; jaw closing, 3; intermixed tongue protrusion, 6) [23]. In an initial dose-finding phase, three subjects were injected with BoNT in prespecified fixed doses [147] assigned to each of the identified target muscles (anterior digastric, genioglossus, masseter, medial pterygoid, and lateral pterygoid muscles). Two of the three patients experienced mild adverse effects even at these doses, so no further dose escalation was performed, and the low dose scheme was used for a subsequent single injection session. All patients received injections tailored to their symptoms using fixed doses of 7.5 to 50 units BoNT in muscles selected from the same set used in phase 1 [147]. Efficacy parameters, including the jaw/tongue portions of the Global Dystonia Severity Rating Scale and Unified Dystonia Rating Scale, showed significant improvements compared to baseline levels at 6 and 12 weeks in unblinded ratings but not in blinded video assessments. Measures of quality of life and speech in addition to the Clinical Global Impression improvement and severity also significantly improved. A total of nine patients experienced mild to moderate side effects, most commonly dysphagia. Five patients received injections for lingual dystonia; four patients developed dysphagia. Subsequent dosing of the genioglossus muscle was decreased from 15 units to 7.5 units [147].

In a retrospective study of 172 patients with lingual dystonia, BoNT was administered to 136 patients, most of whom noted a marked improvement in mastication, pain, and phonation [25]. Transient trouble with swallowing occurred in 12.5% of patients. In another study, 50 units of BoNT injected into each genioglossus muscle was effective in treating lingual dystonia-related tardive dyskinesia [185]. In a study of 30 patients with lingual dystonia who participated in a survey of quality of life evaluated by the OMD questionnaire-25, the average scores dropped from 46.8 to 38.2 after BoNT therapy [182]. Dysphagia occurred in 16.7% of patients. A retrospective chart review reported 17 patients with lingual dystonia, and nine of the patients had received BoNT injections; 55.6% reported symptom improvement, but one patient developed dysphagia requiring gastrostomy tube placement [181].

The comparisons of the total Oromandibular Dystonia Rating Scale scores before and after BoNT therapy in 92 patients revealed a significant decrease after treatment (135.3 vs. 55.2) [3]. Symptoms improved significantly from the baseline to four weeks after BoNT therapy in all the Oromandibular Dystonia Rating Scale subscales, including examiner- (severity, disability, and pain) and patient-rated parameters (general, eating, speech, cosmetic, social/family life, sleep, annoyance, mood, and psychosocial functioning) [3]. Comella [75] performed a systematic review of BoNT in OMD and concluded that BoNT may be the most effective treatment available, improving movement and quality of life in patients with OMD.

### 4.2. Oral Dyskinesia

Little evidence exists suggesting a positive effect of BoNT on classical tardive dyskinesia. In a small single blind study of BoNT therapy [29], patients were videoed at baseline and then prior to and four weeks after three injection sessions. The videos were rated using the Abnormal Involuntary Movement Scale severity scale items 2–4. The patients were injected with BoNT in the orbicularis oris muscle at four sites. Twelve patients participated, but four patients changed their antipsychotic drugs during the study period. When these four patients were included in the analysis, a significant decrease from baseline to the last visit was not seen. However, in an analysis of the remaining eight patients with stable antipsychotic treatment, a significant improvement was seen [29].

### 4.3. Bruxism

The first case report on BoNT therapy for bruxism was published in 1990 [50]. Following the study, positive results of the treatment have been reported [51,206]. Recently, two systematic reviews on the effect of BoNT therapy for bruxism were published [197,207]. One review [207] focused on bite force and EMG activity., and the other reviewed clinical outcomes of BoNT in the management of primary bruxism [197].

In a study by Lee et al. [52], six patients with sleep bruxism were injected with BoNT in both masseter muscles, and six subjects were injected with saline. Nocturnal EMG activity was recorded during sleep from the masseter and temporalis muscles before injection and at 4, 8, and 12 weeks after injection. The frequency of bruxism events in the masseter muscle dropped significantly in the BoNT injection group, while the frequency of the temporalis muscle episodes did not differ between groups. The subjective bruxism symptoms improved in both groups after injection [52].

Sim et al. [54] studied 20 patients with sleep bruxism, where 10 patients received bilateral BoNT injections (25 units per muscle) into the masseter muscles, and the other 10 patients received injections into both the masseter and temporalis muscles. Video–polysomnographic recordings were performed before and at four weeks after injection. BoNT injection did not reduce the frequency, the number of bursts, or the duration of rhythmic masticatory muscle activity episodes in either two groups. However, the injection decreased the peak amplitude of rhythmic masticatory muscle activity EMG burst episodes in both groups. A single BoNT injection was determined to reduce the intensity rather than the frequency of the contraction in jaw closing muscles [54].

Jadhao et al. [55] studied 24 patients with bruxism. The patients were randomly divided into three groups and treated with bilateral injection of BoNT into the masseter and temporalis muscles, saline injections (placebo), or no injections (control). The pain levels at rest and during chewing were assessed, and the occlusal force was measured. All groups were evaluated at the baseline time and at the one-week, three-month, and six-month follow-up visits [55]. The pain at rest and chewing decreased in the BoNT group but persisted unchanged in the placebo and control groups. A significant decrease was seen in the maximal occlusal force in the intervention group compared with the other two groups at three months after treatment. The values for the placebo and control groups did not differ significantly, and after six months, the decrease was significant as compared to the baseline but not when compared to the placebo. These results suggest the efficacy of BoNT for reducing myofascial pain and lowering the bite force [55].

Ondo et al. [56] studied 31 sleep bruxism patients confirmed by polysomnography in a randomized, placebo-controlled trial. A total of 13 participants were injected with 200 units of BoNT (60 units into each masseter and 40 units into each temporalis muscle). The intervention and 10 placebo patients were evaluated 4–8 weeks after the initial visit. Clinical global impression and visual analog scale of change favored the BoNT group, as did the total sleep time and number/duration of bruxing episodes. BoNT effectively and safely improved sleep bruxism [56].

Sim et al. [57] studied 30 subjects with sleep bruxism who were randomly assigned into two groups. The placebo group received saline injections into each masseter muscle, and the intervention group received BoNT injections into each masseter muscle. Audio—video–polysomnographic recordings were taken before, at four weeks after, and at 12 weeks after injection. The peak amplitude of EMG bursts during sleep bruxism showed a significant time and group interaction. The injection decreased the peak amplitude of EMG bursts during sleep bruxism only in the intervention group at 12 weeks. Researchers concluded that a single BoNT injection cannot reduce the genesis of sleep bruxism but can be an effective management option by reducing the strength of the masseter muscle [57].

Sendra et al. [197] reviewed six randomized clinical trials [40,52,53,54,55,56,208] and four case series [209,210,211,212], suggesting that the lack of an established treatment protocol has led to a wide range of BoNT therapy methods. Most studies evaluated the results of a single BoNT administration method without varying the muscles or the injection points. Only one study [54] compared application in the masseter muscles concomitant or not with the temporalis muscle. Redaelli [210] adjusted the BoNT doses according to patient satisfaction. In most studies, the follow-up periods were short. Only one study reported effects maintained for up to one year after treatment [208]. All analyzed studies used questionnaires such as the visual analog scale to evaluate the effects of BoNT. However, such questionnaires are subjective and open to bias. Some studies used objective investigation methods, such as polysomnography [54], EMG [52], digital occlusion [55], and clinical measurements of the maximum mouth opening [212]. No study compared the different brands of BoNT. Therefore, clinical trials comparing different brands are necessary [197].

In most of the above-mentioned studies, the sample size was relatively small, possibly due to the cost of BoNT, the short-term effects, and/or the off-label indication for BoNT injections into the masticatory muscles [197]. All studies demonstrated a positive effect of BoNT on primary bruxism, suggesting BoNT injections to be an effective alternative therapy for the management of primary bruxism [197].

### 4.4. Functional Movement Disorder

Growing evidence, including multiple randomized controlled trials, shows that functional neurological disorders can be effectively treated with psychotherapy and physiotherapy [213,214,215,216,217]. However, BoNT therapy for functional stomatognathic movement disorder is a treatment option for patients with apparent muscle contractions, such as those with OMD [30]. Patients have been treated for symptoms in the orofacial region according to the injection method for OMD based on the clinician experience, without providing any level of evidence [30].

### 4.5. Palatal Tremor

Several case reports and series suggest the effectiveness and safety of BoNT therapy for palatal tremors [59,201,218,219,220,221,222,223]. However, no reports provide conclusive evidence, probably due to the rarity of the condition. Multidiscipline research, including neurology, otorhinolaryngology, and oral and maxillofacial surgery knowledge, will be necessary.

## 5. Practical Guidelines for Treatment

### 5.1. OMD

OMD was not addressed in the 2008 American Academy of Neurology guidelines, and evidence for its treatment with BoNT comes primarily from open-label case series, observational studies, and retrospective chart reviews [224]. In 2016, the American Academy of Neurology published practice guidelines for BoNT administration in the treatment of blepharospasm, cervical dystonia, adult spasticity, and headache [225]. However, OMD was not included in the guideline. In 2011, the European Federation of the Neurological Societies guidelines on the diagnosis and treatment of primary dystonia were published and included recommendations and good practice points regarding BoNT treatment of focal dystonia [226]. The guidelines present level A evidence for the use of BoNT as a first-line treatment for primary cranial or cervical dystonia, but OMD was excluded [163]. A guideline is usually produced after the careful evaluation of previously reported studies by experts in the field. Therefore, the guidelines both in 2008 and 2016 were unable to address OMD.

Although few well-designed trial data exist, BoNT injections are considered the preferred treatment for OMD by several investigators [16,68,69,76]. In 2021, Dressler et al. [155] reported consensus guidelines for BoNT therapy, including general algorithms and dosing tables for dystonia and spasticity. These guidelines also addressed OMD, despite the lack of evidence. However, the authors made no distinction between the lateral pterygoid and the medial pterygoid muscles without EMG guidance, and the present method differs completely from the well-recognized methods used by other OMD specialists [21,49,74,147,173,174].

Treatment algorithms or treatment strategies for OMD have been reported. Sinclair et al. [21] reported a treatment algorithm for OMD, and Bakke et al. [74] presented clinical strategies for BoNT injection in the oromandibular region. More recently, Yoshida reported therapeutic strategies for OMD, including medication, muscle afferent block therapy, splint therapy, BoNT therapy, and surgery [4,173].

Moreover, a clinical practice statement has been released by the American Academy of Oral Medicine, encouraging the identification and referral of patients with suspected OMD to the appropriate oral and/or medical health provider for further evaluation and management [227].

In clinical practice, an experienced clinician would select target muscles and injection sites and determine the dose and allocation for each BoNT injection, corresponding to patient satisfaction and the results of palpation and EMG measurements [4,45,173,174]. The dosing of BoNT should be limited to the necessary minimum amount to minimize the risk of adverse effects and cost and to prevent antibody development. Personalized adjustment of target muscles, sites, and doses will result in much better outcomes than standard methods without individualized planning [4,45,173,174].

### 5.2. Oral Dyskinesia

BoNT therapy for oral dyskinesia is performed according to BoNT therapy methods for OMD. Therefore, a guideline for OMD’s BoNT therapy needs to be developed.

### 5.3. Bruxism

No guideline for BoNT therapy for bruxism exists. Sufficient evidence-level research must be published to prepare guidelines.

### 5.4. Functional Movement Disorder

BoNT therapy is considered a treatment option for functional movement disorders [30] and is administered according to BoNT therapy guidelines for OMD. Therefore, a guideline for BoNT therapy for OMD must be developed.

### 5.5. Palatal Tremor

No practical guideline exists regarding BoNT injection for palatal tremors. The evidence available for this treatment comes primarily from open-label observational studies and retrospective chart reviews [59,201,202,218,219,220,221,222,223]. However, a treatment algorithm for BoNT use in essential palatal tremor was reported based on clinical symptomatology and examination findings and involved muscle groups by a senior author with over 20 years of experience [59]. This algorithm is very useful for BoNT injection in patients.

## 6. Proposals for Research and Future Studies

### 6.1. OMD

In the future, well-designed, randomized, controlled trials with larger sample sizes and longer follow-up periods are required to determine the therapeutic efficacy, optimal dose, duration of effect, adverse effects, brand-specific differences, definite indications, and establishment of a protocol for BoNT therapy. However, the presence of disabilities in patients with OMD places constraints on the traditional placebo–control trial design [23]. As patients seeking with OMD specialists visit from very long distances with very high expectations, making a control group ethically difficult to set [45].

It is important to differentially diagnose patients for BoNT injection. Since BoNT is expensive, it is crucial to predict and exclude non-responders for economic reasons [45]. Discrepant results and adverse effects seen in previous studies may have been related to the injection techniques. Experienced clinicians should inject BoNT at an adequate, personalized dose for each patient. Further, clinicians should comprehensively evaluate disease severity, disability, psychosocial functioning, and impact on the quality of life using the recently developed OMD rating scale [3].

The approaches to OMD treatment should be taken seriously not only by neurologists or neurosurgeons but also by oral and maxillofacial surgeons or dentists [37,38,39,62]. Patients with OMD have been successfully treated with BoNT. However, those with severe trismus related to this disease, or for whom treatment with BoNT injections, muscle afferent block therapy, or sensory trick splint [158] was insufficient, successfully underwent coronoidotomy [160,161,162]. Although a large number of patients worldwide wish to visit our department, only a few can actually make the visit. In reality, it costs an enormous amount of money, and only a few wealthy patients can afford it [97]. Furthermore, overseas travel was prohibited due to coronavirus disease 2019 restrictions, making it impossible to receive medical examinations from abroad. Telemedicine is one of the solutions for treatment accessibility problems [97]. After the direct import and fusion of patient computed tomography scans with a plaster cast model of the maxilla, the optimal needle insertion site over the lateral pterygoid muscle can be determined using software [20]. Such data can be transmitted over the internet from anywhere in the world. As telemedicine for OMD using digital technology in the era of coronavirus disease 2019, computer-aided design and the manufacturing of needle guides for lateral pterygoid muscle injection should be applied in response to the demands of overseas patients with OMD.

### 6.2. Oral Dyskinesia

Most neurologists and psychiatrists believe that oral dyskinesia is incurable; however, numerous cases show that BoNT therapy or dental prosthetic treatment can alleviate the symptoms [193,194]. Therefore, it is important for medical and dental professionals to collaborate in multidisciplinary teams to diagnose and treat patients and improve their quality of life.

### 6.3. Bruxism

Studies with compelling evidence that meet the following requirements are desired to assess the general clinical application of BoNT therapy for bruxism [172]: (1) a definitive diagnosis by polysomnography with video recording or using ambulatory EMG devices; (2) exclusion of other pathological conditions, such as OMD or other neurological diseases; (3) placebo-controlled double-blind comparative studies with a statistically sufficient number of subjects; (4) injection of BoNT by an experienced clinician; (5) adequate dosing; (6) sufficient research period, including follow-up visits; (7) comprehensive evaluation of the changes in symptoms using a validated rating scale; and (8) difference of effect in some brands of BoNT.

### 6.4. Functional Movement Disorder

LaFaver et al. [200] suggested that the lack of physician knowledge, training, and treatment guidelines, along with the limited availability of referral services, are frequent treatment barriers for functional movement disorders.

Kaski et al. [132] suggested that a particular challenge using the comparison of movement-related cortical potentials is distinguishing the line between voluntary and involuntary symptoms. Movement-related cortical potentials slowly increase negative potential shifts 1 to 2 s before voluntary movements and can be observed before mandibular movements [228,229]. Using this technique, some studies have reported the mechanism of dystonia and provided evidence for the dysfunction of basal ganglia in dystonia [230,231,232]. The movement-related cortical potential accompanying jaw movements was reported and showed differences in patients with OMD [233]. The method may be useful to clarify the pathophysiology of the entity and achieve an accurate diagnosis.

Although the study design is clinically difficult, randomized control trials with a large number of patients will be necessary to provide evidence-based therapeutic strategies.

### 6.5. Palatal Tremor

Since palatal tremor is a rare disease, well-designed, multicenter collaborative studies involving neurologists, otorhinolaryngologists, neurosurgeons, and oral surgeons must conduct careful differential diagnosis and the exclusion of functional tremor to obtain reliable data.

## 7. Conclusions

Although previous studies indicated that BoNT therapy is effective and safe with few side effects when properly administered, most of the studies lack conclusive evidence. Neurological knowledge and experience are indispensable to diagnose neurological diseases. While dental knowledge and experience are required for differential diagnoses of dental and oral conditions. Since dentists or dental surgeons are specialists of the stomatognathic system, they are likely to perform more skillful injections into the muscles in the oral region than medical professionals. Every specialist has their strengths. Well-designed multicenter trials with multidisciplinary team approach are necessary to develop treatment plans.

## 8. Methods

This literature review was conducted based on the comprehensive analysis of electronic medical literature databases (PubMed, Scopus, EMBASE, Google scholar, and Japan Medical Abstracts Society) prior to November 30, 2021. Search keywords included oromandibular dystonia, lower cranial dystonia, orofacial dystonia, mandibular dystonia, jaw dystonia, tardive dystonia, lingual dystonia, oral dyskinesia, orolingual dyskinesia, tardive dyskinesia, bruxism, sleep bruxism, awake bruxism, functional (psychogenic) dystonia, functional (psychogenic) movement disorder, palatal tremor, palatal myoclonus, botulinum toxin, and botulinum toxin therapy. Furthermore, a manual search was conducted for sources from articles. No restriction was placed with respect to the original text language.

## Figures and Tables

**Figure 1 toxins-14-00282-f001:**
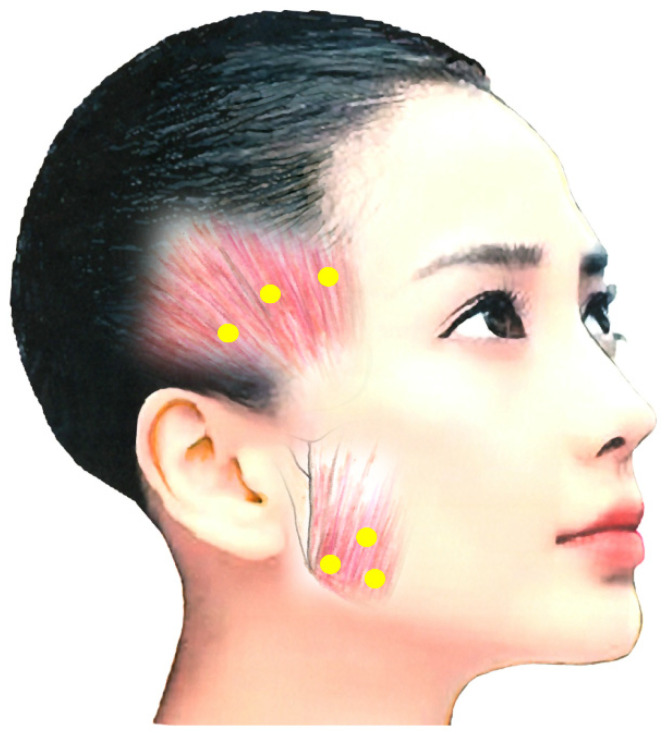
Injection sites for the masseter and temporalis muscles [4,45,173].

**Figure 2 toxins-14-00282-f002:**
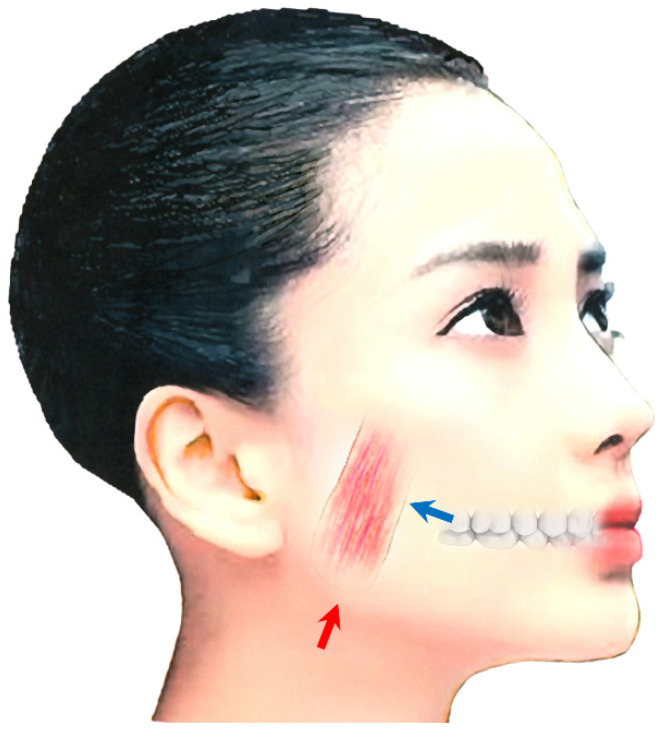
Injection methods for the medial pterygoid muscle; intraoral approach (**blue arrow**) and extraoral oral approach (**red arrow**) [4,22,49,173,174].

**Figure 3 toxins-14-00282-f003:**
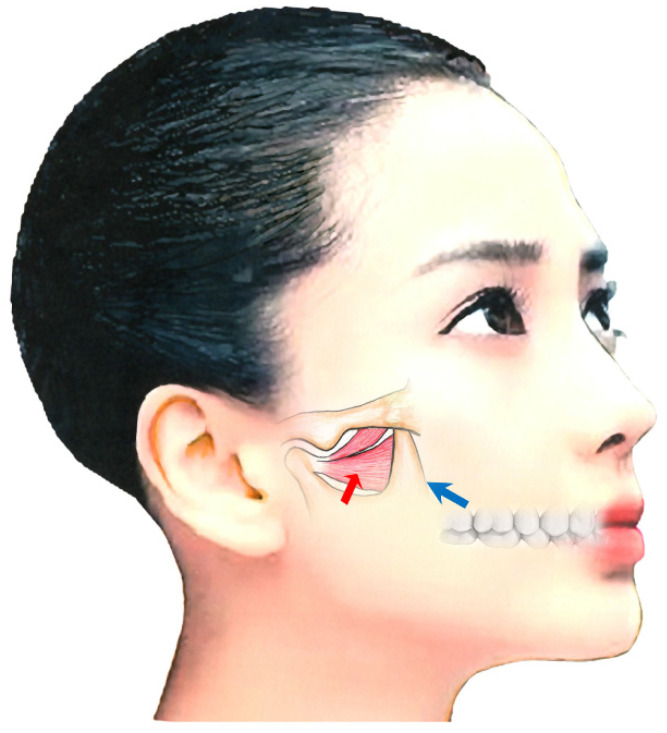
Injection methods for the lateral pterygoid muscle: intraoral approach (**blue arrow**) and extraoral oral approach (**red arrow**) [4,22,49,173,174].

**Figure 4 toxins-14-00282-f004:**
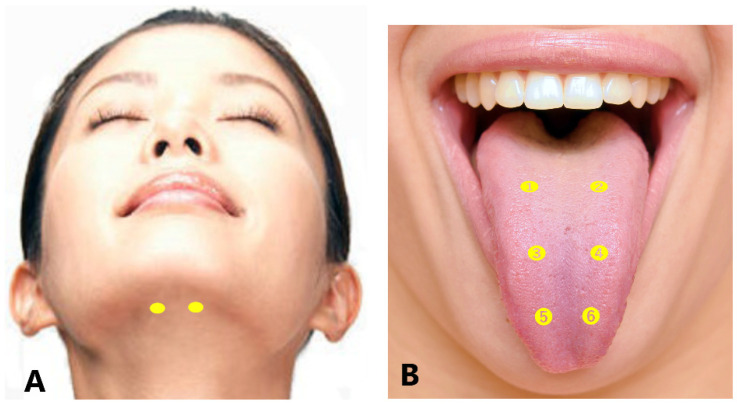
Submandibular sites of BoNT injection for lingual dystonia (**A**). Intraoral sites for BoNT injection (**B**) for protrusion type (➊–➍), laterotrusion type (right deviation (➊, ➌), left deviation (➋, ➍)), and curling type (➊–➏) [4,25,173].

**Table 1 toxins-14-00282-t001:** Recommended target muscles and doses of BoNT (onabotulinumtoxinA: Botox®) for jaw closing dystonia [4,45,173].

Main Muscles (Doses (Units))	Additional Muscles (Doses (Units))	
Bilateral masseter (10–50)	Bilateral medial pterygoid (10–30)	Insufficient cases only for masseter and temporalis, or cases in which the effect was diminished by repeated injections
Bilateral temporalis (10–50)	Contralateral or bilateral lateral pterygoid (10–30)	With mandibular deviation, grinding, and myalgia of the lateral pterygoid muscle

onabotulinumtoxinA:incobotulinumtoxinA = 1:1; onabotulinumtoxinA:abobotulinumtoxinA = 1:2.5 [13,164].

**Table 2 toxins-14-00282-t002:** Recommended muscles and doses of BoNT (onabotulinumtoxinA: Botox®) for jaw opening, deviation, and protrusion dystonia [4,22,49,173,174].

Subtypes	Main Muscles (Doses (Units))	Additional Muscles (Doses (Units))	
Jaw opening dystonia	Bilateral lateral pterygoid (10–50)	Anterior digastric (5–10)	Insufficient cases only for lateral pterygoid
		Platysma (10–20)	With anterior neck tension
		Genioglossus (10–20)	With tongue protrusion
Jaw deviation dystonia	Contralateral lateral pterygoid (10–50)	Ipsilateral posterior temporalis (10–20)	Insufficient cases only for lateral pterygoid
		Platysma (10–20)	With anterior neck tension
Jaw protrusion dystonia	Bilateral lateral pterygoid (10–40)	Anterior digastric (5–10)	Insufficient cases only for lateral pterygoid
		Platysma (10–20)	With anterior neck tension

onabotulinumtoxinA:incobotulinumtoxinA = 1:1; onabotulinumtoxinA:abobotulinumtoxinA = 1:2.5 [13,164].

**Table 3 toxins-14-00282-t003:** Recommended muscles and doses of BoNT (onabotulinumtoxinA: Botox®) for lingual dystonia [4,25,173].

Subtypes	Doses (Units)	Main Muscles (Doses (Units))	Additional Muscles	
**Protrusion type**	15–60	Bilateral genioglossus (50–100% of total dose)	Ipsilateral superior and inferior longitudinal (0–50%)	With laterotrusion
			Bilateral superior longitudinal (0–50%)	With curling
			Bilateral vertical (0–50%)	With flattening
			Bilateral transverse (0–50%)	With elongation
			Bilateral lateral pterygoid (0–50%)	With jaw opening
Retraction type	15–50	Bilateral genioglossus (30–70% of total dose)	Intrinsic and geniohyoid (30–70%)	Insufficient cases only for genioglossus
Laterotrusion type	10–40	Ipsilateral superior and inferior longitudinal (70–100% of total dose)	Contralateral genioglossus (0–30%)	Insufficient cases only for superior and inferior longitudinal
Curling type	10–40	Bilateral superior longitudinal (80–100% of total dose)	Bilateral genioglossus (0–20%)	With protrusion

onabotulinumtoxinA:incobotulinumtoxinA = 1:1; onabotulinumtoxinA: abobotulinumtoxinA = 1:2.5 [13,164].

## Data Availability

The raw data supporting the conclusions of this article will be made available by the authors, without undue reservation, to any qualified researcher.

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
