# Peer review of "Botulinum Toxin Therapy for Oromandibular Dystonia and Other Movement Disorders in the Stomatognathic System"

_toxins, 2022, doi:10.3390/toxins14040282_

Round 1
Reviewer 1 Report
Dear authors, this is an outstanding and comprehensive review on the topic of BoNT-Treatment for a rare neurological disease that falls out of scope for most users of BoNT, but still occurs regularly in the large specialist centers, so the wealth of information included is highly relevant for the care of many.
I only have few comments:
- Overall I liked how the article is structured and the flow of thoughts. At the end, it feels as if the "fuel" did run out ...
- The review has its strengths when it comes to the dental surgeons's perspective from which it is written. It lacks the perspective of the neurologist. E.g. on page 22 the author mentions the use of ERCP for the distinction of functional symptoms. While it is true that this technology exists, it is not used routinely in the clinical setting in any Botulinumtoxin clinic I currently am aware of.
- Sonography which emerges as a critical tool to inject ultraprecisely , e.g. in genioglossus, omohyoideus, digrastricus, infrahyoidal muscles is non-existent in the paper - this is not state-of-the-art.
- There is not a lot of cosmetic expertise when the authors do not differentiate injections of the lips that produce dramatically different results if fibres close to orofice (increases lip pursing, decreases pressing) or distant of orofice (vv) are injected.
- The paper very often tries to be extremely precise, e.g. by mentioning angles and depths, but these are very theoretical numbers that would need visulalizations to be of use to the reader - none are provided.
- on page 11 two refs of "early studies" are mentioned for serious complications, but only 164 is "early". This topic (life threatening AE following BoNT injections) has suffered from ref-copying already a lot so more precision would be adequate.
- If authors mention preparations names ,e.g. Botox(r) (see Table headers 1-3) they should use the trademark or register sign - or better: omit the "product placement".
Altogher, this is a very comprehensive paper - but to me if felt overwhelming with its 23 pages of text -more like a textbook than a paper actually.
I would suggest the following
- Rename to make the dental surgeon's perspective more transparent
- more focus - more paper like - refrain from neurological topics such as ERCP - no retelling of other authors work
- graphical elements would make a huge difference
- Tabular lists of Literature and main results would give faster and better overview.
Author Response
Dear authors, this is an outstanding and comprehensive review on the topic of BoNT-Treatment for a rare neurological disease that falls out of scope for most users of BoNT, but still occurs regularly in the large specialist centers, so the wealth of information included is highly relevant for the care of many.
Thank you very much for carefully reviewing my manuscript and valuable comments.
Overall I liked how the article is structured and the flow of thoughts. At the end, it feels as if the "fuel" did run out ...
The manuscript became very long, took long time to write and I was really exhausted. As you pointed out, there is a feeling that it has tailed away. Furthermore, unfortunately there are very little evidence in the studies of OMD or other movement disorders in this region. There were almost no evidence-based studies or consensus guidelines to report in this article. Currently, I am planning RCT for OMD and later other diseases. I hope reliable guidelines or a therapeutic strategy based on evidence just like those for cervical dystonia or blepharospasm will be established in the near future.
The review has its strengths when it comes to the dental surgeons's perspective from which it is written. It lacks the perspective of the neurologist. E.g. on page 22 the author mentions the use of ERCP for the distinction of functional symptoms. While it is true that this technology exists, it is not used routinely in the clinical setting in any Botulinumtoxin clinic I currently am aware of.
Over 20 years ago, I was doing research and clinical practice in the Department of Neurology. At that time I studied MRCP (movement-related cortical potential) and reported on significant differences between patients with OMD and normal controls. I think neurophysiology of functional movement disorders remains unclear. In particular, no studies have been conducted on this condition in the stomatognathic region. Indeed, MRCP is not a routine test in the movement disorder clinic, however I think it makes sense in terms of future studies to clarify the pathophysiology of patients with functional movement disorders in oral region. I explained the above and cited some references.
I revised the sentences as follows;
“Kaski et al. [132] suggested that a particular challenge using the comparison of movement-related cortical potentials is distinguishing the line between voluntary and involuntary symptoms. Movement-related cortical potentials slowly increase negative potential shifts 1-2 s before voluntary movements and can be observed before mandibular movements [226,227]. Using this technique, some studies have reported the mechanism of dystonia and provided evidence for the dysfunction of basal ganglia in dystonia [228–230]. The movement-related cortical potentials accompanying jaw movements were reported and showed differences in patients with OMD [231]. The method may be useful to clarify pathophysiology of the entity and achieve an accurate diagnosis.”
Sonography which emerges as a critical tool to inject ultraprecisely , e.g. in genioglossus, omohyoideus, digrastricus, infrahyoidal muscles is non-existent in the paper - this is not state-of-the-art.
Thank you for important comment. I thought I should mention it, but I forgot to mention it. I added sentences on sonography as a critical tool. I added following sentences and references on sonography for botulinum toxin therapy.
“Sonography is important for identifying target muscles and preventing damage to other tissues and is often used for cervical dystonia [170,171]. This method is also very useful for OMD and is recommended for use in combination with other methods such as EMG.”
There is not a lot of cosmetic expertise when the authors do not differentiate injections of the lips that produce dramatically different results if fibres close to orofice (increases lip pursing, decreases pressing) or distant of orofice (vv) are injected.
I added following sentence and cited a reference.
“Since the fibers of the orbicularis oris muscle close to the orifice are responsible for pursing the lips, and the fibers distant from the orifice press the lips against the teeth [149], the dose and site should be adjusted according to the symptoms [4].”
The paper very often tries to be extremely precise, e.g. by mentioning angles and depths, but these are very theoretical numbers that would need visulalizations to be of use to the reader - none are provided.
I added four figures. I think it's easier for the reader to visually understand the methods.
on page 11 two refs of "early studies" are mentioned for serious complications, but only 164 is "early". This topic (life threatening AE following BoNT injections) has suffered from ref-copying already a lot so more precision would be adequate.
I changed the word "early" to "previous".
If authors mention preparations names ,e.g. Botox(r) (see Table headers 1-3) they should use the trademark or register sign - or better: omit the "product placement".
I used the trademark in the Tables.
Altogher, this is a very comprehensive paper - but to me if felt overwhelming with its 23 pages of text -more like a textbook than a paper actually.
Indeed, the manuscript has become too long. Initially, I planned to include hemifacial spasm and facial synkinesis, but I had to omit it because it became too long. Fortunately, Toxins has no word limit.
Rename to make the dental surgeon's perspective more transparent
This article is a narrative review from the perspective of an oral surgeon. I added the words “from the perspective of an oral surgeon” in the Introduction.
more focus - more paper like - refrain from neurological topics such as ERCP - no retelling of other authors work
I think neurophysiology of functional movement disorders remains unclear. In particular, no studies have been conducted on functional movement disorders in the stomatognathic region. Indeed, MRCP is not a routine test in the movement disorder clinic, however I think it makes sense in terms of future studies to clarify the pathophysiology of patients with functional movement disorders in oral region.
graphical elements would make a huge difference
I added figures to make it more visually understandable.
Tabular lists of Literature and main results would give faster and better overview.
You are right on that point. I attempted to make tabular lists of literature like other reviews. For example, a review by Anandan & Jankovic published in Toxins is very excellent, however the characters in the tabular lists are too little to read (Font size is only 7). Therefore, I decided to write the results in the text for better readability.

Reviewer 2 Report
Dear Authors,
With great pleasure I reviewed the manuscript entitled "Botulinum Toxin Therapy for Oromandibular Dystonia and Other Movement Disorders in the Stomatognathic System", submitted to Toxins. It is clear that there has been put a lot of effort in this study. The authors made their attempt to review previous studies on botulinum toxin therapy in various movement disorders. In addition, they provide thorough background information about these disorders. I have, however, one aspect that needs to be discussed. The strict guidelines for performing a systematic review were not followed, which can raise questions about its quality and trustworthiness. So I guess that this is a scoping review of the literature? Or perhaps a integrative or critical review approach?
Author Response
Dear Authors,
With great pleasure I reviewed the manuscript entitled "Botulinum Toxin Therapy for Oromandibular Dystonia and Other Movement Disorders in the Stomatognathic System", submitted to Toxins. It is clear that there has been put a lot of effort in this study. The authors made their attempt to review previous studies on botulinum toxin therapy in various movement disorders. In addition, they provide thorough background information about these disorders. I have, however, one aspect that needs to be discussed. The strict guidelines for performing a systematic review were not followed, which can raise questions about its quality and trustworthiness. So I guess that this is a scoping review of the literature? Or perhaps a integrative or critical review approach?
Thank you very much for reviewing my manuscript and your comment. This paper is not an systematic review, but rather a narrative review from the perspective of an oral surgeon. Therefore, I’m afraid that the figure of PRISMA is not necessary. I added the words “from the perspective of an oral surgeon” in the Introduction and some details in the Method.

Reviewer 3 Report
In this manuscript, the authors reviewed the clinical problems, treatment challenges, and proposal for future studies of BoNT therapy for oromandibular dystonia, oral dyskinesia, bruxism, functional stomatognathic movement disorder (functional movement disorders in the stomatognathic system), and palatal tremor.
This manuscript is interesting und well-written; unfortunately, this manuscript needs substantial improvements and corrections before publishing may be possible. It may sound a bit astonishing, but the main point of my criticism is that - for an even more comprehensive review - more references are need to get a really good overview of the field.
General points:
Please add a list of abbreviations before References section to your manuscript.
For better readability, please add at least two Figures to your manuscript.
Special points:
This manuscript should be substantially improved, i. e., by substantial references in the field:
- Introduction
Lines 21-33: please add multiple references at the end of each of these sentences.
Lines 38-40: please add multiple references at the end of this sentence.
- The Clinical Problem: History, Presentation, and Epidemiology
Lines 52-54: please add more references at the end of this sentence.
Lines 136-137: please add more references at the end of this sentence.
Lines 139-146: please add multiple references at the end of each of these sentences.
Lines 168-169: please add more references at the end of this sentence.
Lines 211-212: please add multiple references at the end of each of these sentences.
Lines 218-224: please add multiple references at the end of each of these sentences.
Lines 226-231: please add multiple references at the end of each of these sentences.
Lines 241-242: please add more references at the end of this sentence.
Lines 251-252: please add multiple references at the end of this sentence.
Lines 310-312: please add multiple references at the end of this sentence.
Lines 337-340: please add more references at the end of this sentence.
Lines 340-345: please add multiple references at the end of each of these sentences.
Lines 357-358: please add multiple references at the end of this sentence.
Lines 358-365: please add more references at the end of each of these sentences.
Lines 368-371: please add multiple references at the end of each of these sentences.
Lines 376-378: please add multiple references at the end of this sentence.
Lines 381-383: please add multiple references at the end of each of these sentences.
Lines 393-394: please add multiple references at the end of this sentence.
Lines 403-405: please add multiple references at the end of this sentence.
Table 1: please add also the references and references numbers according to your List of references for each use and each dose.
Lines 413-415: please add multiple references at the end of each of these sentences.
Lines 418-420: please add multiple references at the end of each of these sentences.
Lines 426-428: please add multiple references at the end of each of these sentences.
Lines 437-439: please add multiple references at the end of each of these sentences.
Table 2: please add also the references and references numbers according to your List of references for each use and each dose.
Lines 465-467: please add multiple references at the end of each of these sentences.
Lines 473-474: please add multiple references at the end of this sentence.
Lines 500-507: please add multiple references at the end of each of these sentences.
Lines 511-514: please add multiple references at the end of each of these sentences.
Lines 520-522: please add multiple references at the end of each of these sentences.
Table3: please add also the references and references numbers according to your List of references for each use and each dose.
Line 541: please add multiple references at the end of this sentence.
Lines 546-547: please add multiple references at the end of each of these sentences.
Lines 553-554: please add multiple references at the end of each of these sentences.
Lines 580-584: please add multiple references at the end of each of these sentences.
Line 591: please add multiple references at the end of this sentence.
Line 602: please add multiple references at the end of this sentence.
Lines 609-611: please add multiple references at the end of this sentence.
Lines 623-629: please add multiple references at the end of each of these sentences.
Lines 649-655: please add multiple references at the end of each of these sentences.
Lines 659-664: please add multiple references at the end of each of these sentences.
Lines 665-666: please add multiple references at the end of this sentence.
Lines 669-674: please add multiple references at the end of each of these sentences.
Lines 678-681: please add multiple references at the end of each of these sentences.
Lines 685-689: please add multiple references at the end of each of these sentences.
Lines 695-701: please add multiple references at the end of each of these sentences.
Lines 723-728: please add multiple references at the end of each of these sentences.
Lines 734-735: please add multiple references at the end of this sentence.
Lines 742-745: please add multiple references at the end of this sentence.
Lines 768-770: please add multiple references at the end of this sentence.
Lines 859-860: please add multiple references at the end of this sentence.
Lines 987-988: please add multiple references at the end of this sentence.
Lines 993-995: please add multiple references at the end of this sentence.
Lines 1039-1041: please add multiple references at the end of this sentence.
Methods
Lines 1085-1092: please add also the exact time period, in that you made your literature searches of comprehensive electronic medical literature databases (PubMed, Scopus, EMBASE, Google scholar, Japan Medical Abstracts Society).
Author Response
This manuscript is interesting und well-written; unfortunately, this manuscript needs substantial improvements and corrections before publishing may be possible. It may sound a bit astonishing, but the main point of my criticism is that - for an even more comprehensive review - more references are need to get a really good overview of the field.
Thank you very much for carefully reviewing my manuscript and valuable comments.
General points:
Please add a list of abbreviations before References section to your manuscript.
I added a list of abbreviations.
BoNT: botulinum neurotoxin
OMD: oromandibular dystonia
EMG: electromyography
For better readability, please add at least two Figures to your manuscript.
I added four Figures for better readability.
Special points:
This manuscript should be substantially improved, i. e., by substantial references in the field:
- Introduction
Lines 21-33: please add multiple references at the end of each of these sentences.
Lines 38-40: please add multiple references at the end of this sentence.
Thank you for very carefully checking my manuscript.
I added multiple references at the end of the sentences.
- The Clinical Problem: History, Presentation, and Epidemiology
Lines 52-54: please add more references at the end of this sentence.
Lines 136-137: please add more references at the end of this sentence.
Lines 139-146: please add multiple references at the end of each of these sentences.
Lines 168-169: please add more references at the end of this sentence.
Lines 211-212: please add multiple references at the end of each of these sentences.
Lines 218-224: please add multiple references at the end of each of these sentences.
Lines 226-231: please add multiple references at the end of each of these sentences.
Lines 241-242: please add more references at the end of this sentence.
Lines 251-252: please add multiple references at the end of this sentence.
Lines 310-312: please add multiple references at the end of this sentence.
Lines 337-340: please add more references at the end of this sentence.
Lines 340-345: please add multiple references at the end of each of these sentences.
Lines 357-358: please add multiple references at the end of this sentence.
Lines 358-365: please add more references at the end of each of these sentences.
Lines 368-371: please add multiple references at the end of each of these sentences.
Lines 376-378: please add multiple references at the end of this sentence.
Lines 381-383: please add multiple references at the end of each of these sentences.
Lines 393-394: please add multiple references at the end of this sentence.
Lines 403-405: please add multiple references at the end of this sentence.
I added multiple references at the end of the sentences.
Table 1: please add also the references and references numbers according to your List of references for each use and each dose.
I added multiple references at the end of the sentences.
Lines 413-415: please add multiple references at the end of each of these sentences.
Lines 418-420: please add multiple references at the end of each of these sentences.
Lines 426-428: please add multiple references at the end of each of these sentences.
Lines 437-439: please add multiple references at the end of each of these sentences.
Table 2: please add also the references and references numbers according to your List of references for each use and each dose.
I added multiple references at the end of the sentences.
Lines 465-467: please add multiple references at the end of each of these sentences.
Lines 473-474: please add multiple references at the end of this sentence.
Lines 500-507: please add multiple references at the end of each of these sentences.
Lines 511-514: please add multiple references at the end of each of these sentences.
Lines 520-522: please add multiple references at the end of each of these sentences.
I added multiple references at the end of the sentences.
Table3: please add also the references and references numbers according to your List of references for each use and each dose.
I added multiple references at the end of the sentences.
Line 541: please add multiple references at the end of this sentence.
Lines 546-547: please add multiple references at the end of each of these sentences.
Lines 553-554: please add multiple references at the end of each of these sentences.
Lines 580-584: please add multiple references at the end of each of these sentences.
Line 591: please add multiple references at the end of this sentence.
Line 602: please add multiple references at the end of this sentence.
Lines 609-611: please add multiple references at the end of this sentence.
Lines 623-629: please add multiple references at the end of each of these sentences.
Lines 649-655: please add multiple references at the end of each of these sentences.
Lines 659-664: please add multiple references at the end of each of these sentences.
Lines 665-666: please add multiple references at the end of this sentence.
Lines 669-674: please add multiple references at the end of each of these sentences.
Lines 678-681: please add multiple references at the end of each of these sentences.
Lines 685-689: please add multiple references at the end of each of these sentences.
Lines 695-701: please add multiple references at the end of each of these sentences.
Lines 723-728: please add multiple references at the end of each of these sentences.
Lines 734-735: please add multiple references at the end of this sentence.
Lines 742-745: please add multiple references at the end of this sentence.
Lines 768-770: please add multiple references at the end of this sentence.
Lines 859-860: please add multiple references at the end of this sentence.
Lines 987-988: please add multiple references at the end of this sentence.
Lines 993-995: please add multiple references at the end of this sentence.
Lines 1039-1041: please add multiple references at the end of this sentence.
I added multiple references at the end of the sentences.
Methods
Lines 1085-1092: please add also the exact time period, in that you made your literature searches of comprehensive electronic medical literature databases (PubMed, Scopus, EMBASE, Google scholar, Japan Medical Abstracts Society).
I added necessary detail (time period) in the Method.

Reviewer 4 Report
An interest, exhaustive, and well-written manuscript. I have several suggestions to improve its quality:
1) The term "dysphasia" should be corrected to "dysphagia" throughout the text.
2) Orofacial dyskinesia is not only due to neuroleptic drugs. Several other drugs can cause it (see Ortí-Pareja et al. Drug-induced tardive syndromes. Parkinsonism Relat Disord. 1999 Apr;5(1-2):59-65).
3) Choreatic orofacial dyskinesia usually show worsening with anticholinergics, while dystonia and rabbit syndrome improve.
4) In page 18 [Bakke] should be expressed as a numeric reference.
5) The reference Alonso-Navarro H et al. [Treatment of severe bruxism with botulinum toxin type A]. Rev Neurol. 2011 Jul 16;53(2):73-6, should be included. This paper reported the outcome of 19 patients with severe bruxism who underwent periodical treatment with botulinum toxin A infiltrations in both temporal and masseter muscles, using initial doses of 25 IU per muscle, during a follow-up period ranging from 0.5 to 11 years, with adjusting the dosis, according to the response degree, in the follow-up visits, Final dosis required and duration of the effect were also calculated.
6) A detailed description of the search strategy, including a Figure with the PRISMA flowchart, should be appropriate.
Author Response
An interest, exhaustive, and well-written manuscript. I have several suggestions to improve its quality:
Thank you very much for reviewing my manuscript and valuable comments.
1) The term "dysphasia" should be corrected to "dysphagia" throughout the text.
Thank you for carefully checking my manuscript. I did not notice it at all. I corrected the term.
2) Orofacial dyskinesia is not only due to neuroleptic drugs. Several other drugs can cause it (see Ortí-Pareja et al. Drug-induced tardive syndromes. Parkinsonism Relat Disord. 1999 Apr;5(1-2):59-65).
Thank you for valuable comment. I added sentences and the reference.
3) Choreatic orofacial dyskinesia usually show worsening with anticholinergics, while dystonia and rabbit syndrome improve.
I added the following sentence in the text.
“However, the symptoms of patients with oral dyskinesia often worsen with anticholinergics. Further, elderly patients require careful monitoring of symptoms, such as dementia and constipation caused by the anticholinergic effect of this drug.”
4) In page 18 [Bakke] should be expressed as a numeric reference.
I added the study by Bakke et al.
5) The reference Alonso-Navarro H et al. [Treatment of severe bruxism with botulinum toxin type A]. Rev Neurol. 2011 Jul 16;53(2):73-6, should be included. This paper reported the outcome of 19 patients with severe bruxism who underwent periodical treatment with botulinum toxin A infiltrations in both temporal and masseter muscles, using initial doses of 25 IU per muscle, during a follow-up period ranging from 0.5 to 11 years, with adjusting the dosis, according to the response degree, in the follow-up visits, Final dosis required and duration of the effect were also calculated.
Perhaps due to Spanish paper, I had not noticed the study. I cited the study.
6) A detailed description of the search strategy, including a Figure with the PRISMA flowchart, should be appropriate.
This paper is not an systematic review, but rather a narrative review. Therefore, I’m afraid that the figure of PRISMA is not necessary. I added some details in the Method.

Round 2
Reviewer 1 Report
Dear authors,
thank you for adapting the paper according to the reviewers suggestions. The paper can proceed to publication.
KR
Author Response
Dear authors,
thank you for adapting the paper according to the reviewers suggestions. The paper can proceed to publication.
Thank you very much for carefully reviewing my manuscript and valuable comments. Thanks to your comments, I think my manuscript has improved.

Reviewer 3 Report
Dear authors, thank you for your corrections. The manuscript was intensively improved. Unfortunately, the authors did not react on all my previous proposals, and some problems with the manuscript appeared during their corrections. Therefore, the manuscript still needs further improvements and corrections before publishing may be possible.
General points:
Once again, please add a list of abbreviations before the References section to your manuscript. Please, also include all abbreviations from your manuscript to the list of abbreviations.
Once again, please add a “Future perspectives” section to your manuscript.
Special points:
Once again, this manuscript should be further improved, i. e., by substantial references in the field:
- Introduction
Lines 22-28: please add multiple references at the end of each of these sentences.
- The Clinical Problem: History, Presentation, and Epidemiology
Lines 53-58: please add more references at the end of this sentence.
Lines 182-183: please add more references at the end of this sentence.
Lines 216-217: please add more references at the end of this sentence.
Lines 249-250: please add more references at the end of this sentence.
Lines 344-347: please add more references at the end of this sentence.
Lines 528-529: please add more references at the end of this sentence.
Lines 538-541: please add more references at the end of this sentence.
Line 615: please add more references at the end of this sentence.
Line 629: please add more references at the end of this sentence.
Lines 726-728: please add more references at the end of this sentence.
Lines 1041-1042: please add more references at the end of this sentence.
Lines 1045-1047: please add more references at the end of this sentence.
Figures Legends
Figure 1 Legend: please add appropriate references for these injection points like you did in the Legend of Figure 4.
Figure 2 Legend: please add appropriate references for these injection points like you did in the Legend of Figure 4.
Figure 3 Legend: please add appropriate references for these injection points like you did in the Legend of Figure 4.
Author Response
Dear authors, thank you for your corrections. The manuscript was intensively improved. Unfortunately, the authors did not react on all my previous proposals, and some problems with the manuscript appeared during their corrections. Therefore, the manuscript still needs further improvements and corrections before publishing may be possible.
Thank you for very carefully checking my manuscript. I revised my manuscript. Thanks to your comments, I think my manuscript has improved.
General points:
Once again, please add a list of abbreviations before the References section to your manuscript. Please, also include all abbreviations from your manuscript to the list of abbreviations.
I had added a list of abbreviations to my response to reviewers, but forgot to add them to the text. I added the list before the References.
Abbreviations
BoNT: botulinum neurotoxin
OMD: oromandibular dystonia
EMG: electromyography
Once again, please add a “Future perspectives” section to your manuscript.
This manuscript is written for a Special Issue: Botulinum Toxin in the Movement Disorders Clinic: State of the Art. This special issue has a fixed format (1. Introduction, 2. The Clinical Problem: History, Presentation, and Epidemiology, 3. Treatment Challenges and Pitfalls, 4. Reported Trials – Evidence Based Medicine, 5. Practical Guidelines for Treatment, 6. Proposals for Research and Future Studies), and I cannot change it on my own. I wrote "Future perspectives" for each disease in 6. “Proposals for Research and Future Studies”.
Special points:
Once again, this manuscript should be further improved, i. e., by substantial references in the field:
- Introduction
Lines 22-28: please add multiple references at the end of each of these sentences.
The Key Contribution of this review was written in lines 22-25, so I don't think I should cite the references. Two references were cited in lines 26-28.
- The Clinical Problem: History, Presentation, and Epidemiology
Lines 53-58: please add more references at the end of this sentence.
I think that the citations should be kept to the minimum necessary representative references in the Introduction, as each disease and previous reports will be explained later in detail.
Lines 182-183: please add more references at the end of this sentence.
I added multiple references at the end of the sentence.
Lines 216-217: please add more references at the end of this sentence.
Multiple citations are not possible because the citation is based on the results from one study.
Lines 249-250: please add more references at the end of this sentence.
I cited the International consensus published in one paper. Therefore, multiple citations are not possible.
Lines 344-347: please add more references at the end of this sentence.
I cited the results published in two studies. Therefore, multiple citations are not possible.
Lines 528-529: please add more references at the end of this sentence.
I added appropriate references.
Lines 538-541: please add more references at the end of this sentence.
I added multiple references.
Line 615: please add more references at the end of this sentence.
I added a new citation on anatomy (Berkovitz, B.K.B. Tongue. In Gray’s Anatomy, 41st ed.; Standring, S., Ed.; Elsevier: Amsterdam, The Netherlands, 2016; pp. 511–517.).
Line 629: please add more references at the end of this sentence.
I added multiple references.
Lines 726-728: please add more references at the end of this sentence.
I cited results of one paper. Therefore, more citations are not possible.
Lines 1041-1042: please add more references at the end of this sentence.
I cited results of one paper. Therefore, more citations are not possible.
Lines 1045-1047: please add more references at the end of this sentence.
I cited 4 references.
Figures Legends
Figure 1 Legend: please add appropriate references for these injection points like you did in the Legend of Figure 4.
Figure 2 Legend: please add appropriate references for these injection points like you did in the Legend of Figure 4.
Figure 3 Legend: please add appropriate references for these injection points like you did in the Legend of Figure 4.
I added appropriate references in Legends of Figures 1-3.

Reviewer 4 Report
No additional comments
Author Response
No additional comments
Thank you very much for reviewing my manuscript and valuable comments. Thanks to your comments, I think my manuscript has improved.

Round 3
Reviewer 3 Report
The authors fulfilled my comments. Now, the manuscript is fine.